# Improved Variance-Aware Confidence Sets for Linear Bandits and Linear Mixture MDP

**Zihan Zhang**[*]
Tsinghua University
zihan-zh17@mails.tsinghua.edu.cn

**Jiaqi Yang**[*]
Tsinghua University
yangjq17@gmail.com

**Xiangyang Ji**
Tsinghua University
xyji@tsinghua.edu.cn

**Simon S. Du**
University of Washington
ssdu@cs.washington.edu

## Abstract

This paper presents new *variance-aware* confidence sets for linear bandits and linear mixture Markov Decision Processes (MDPs). With the new confidence sets, we obtain the follow regret bounds:

- For linear bandits, we obtain an $\widetilde{O}(\text{poly}(d)\sqrt{1 + \sum_{k=1}^{K} \sigma_k^2})$ data-dependent regret bound, where $d$ is the feature dimension, $K$ is the number of rounds, and $\sigma_k^2$ is the *unknown* variance of the reward at the $k$-th round. This is the first regret bound that only scales with the variance and the dimension but *no explicit polynomial dependency on $K$*. When variances are small, this bound can be significantly smaller than the $\widetilde{\Theta}\left(d\sqrt{K}\right)$ worst-case regret bound.

- For linear mixture MDPs, we obtain an $\widetilde{O}(\text{poly}(d, \log H)\sqrt{K})$ regret bound, where $d$ is the number of base models, $K$ is the number of episodes, and $H$ is the planning horizon. This is the first regret bound that only scales *logarithmically* with $H$ in the reinforcement learning with linear function approximation setting, thus *exponentially improving* existing results, and resolving an open problem in [Zhou et al., 2020a].

We develop three technical ideas that may be of independent interest: 1) applications of the peeling technique to both the input norm and the variance magnitude, 2) a recursion-based estimator for the variance, and 3) a new convex potential lemma that generalizes the seminal elliptical potential lemma.

## 1 Introduction

In sequential decision-making problems such as bandits and reinforcement learning (RL), the agent chooses an action based on the current state, with the goal to maximize the total reward. When the state-action space is large, function approximation is often used for generalization. One of the most fundamental and widely used methods is linear function approximation.

For (infinite-actioned) linear bandits, the minimax-optimal regret bound is $\widetilde{\Theta}(d\sqrt{K})$ [Dani et al., 2008, Abbasi-Yadkori et al., 2011], where $d$ is the feature dimension and $K$ is the number of total rounds played by the agent.[2] However, oftentimes the worst-case analysis is overly pessimistic, and

---

[*]Equal contribution.

[2]We follow the reinforcement learning convention to use $K$ to denote the total number of rounds / episodes.

35th Conference on Neural Information Processing Systems (NeurIPS 2021).

it is possible to obtain data-dependent bound that is substantially smaller than $\widetilde{O}(d\sqrt{K})$ in benign scenarios.

One direction to study is the variance magnitude. As a motivating example, in linear bandits, if there is no noise (variance is $0$), one only needs to pay at most $d$ regret to identify the best action because $d$ samples are sufficient to recover the underlying linear coefficients (in general position). This constant-type regret bound is much smaller than the $\sqrt{K}$-type regret bound in the worst case where the variance magnitude is a lower bounded constant. Therefore, a natural question is:

> **Can we design an algorithm that adapts to the variance magnitude, and its regret degrades gracefully from the benign noiseless constant-type bound to the worst-case $\sqrt{K}$-type bound?**

In RL, exploiting the variance information is also important. For tabular RL, one needs to utilize the variance information, e.g., Bernstein-type exploration bonus to achieve the minimax optimal regret [Azar et al., 2017, Zanette and Brunskill, 2019, Zhang et al., 2020c,a, Menard et al., 2021, Dann et al., 2019]. For example, the recently proposed MVP algorithm [Zhang et al., 2020a], enjoys an $\widetilde{O}(\mathrm{polylog}(H) \times (\sqrt{SAK} + S^2 A))$ regret bound, where $S$ is the number of states, $A$ is the number of actions, $H$ is the planning horizon, and $K$ is the total number of episodes. [3][4] Notably, this regret bound only scales *logarithmically* with $H$. On the other hand, without using the variance information, e.g., using Hoeffding-type bonus instead of Bernstein-type bonus, algorithms would suffer a regret that scales *polynomially* with $H$ [Azar et al., 2017].

Going beyond tabular RL, a recent line of work studied RL with linear function approximation with different assumptions [Yang and Wang, 2019, Modi et al., 2020, Jin et al., 2020, Ayoub et al., 2020, Zhou et al., 2020a, Modi et al., 2020]. Our paper studies the linear mixture Markov Decision Process (MDP) setting [Modi et al., 2020, Ayoub et al., 2020, Zhou et al., 2020a], where the transition probability can be represented by a linear function of some features or base models. This model-based assumption is motivated by problems in robotics and queuing systems. We refer readers to Ayoub et al. [2020] for more discussions.

For this linear mixture MDP setting, previous works can obtain regret bounds in the form $\widetilde{O}(\mathrm{poly}(d, H)\sqrt{K})$, where $d$ is the number of base models. While these bounds do not scale with $SA$, they scale *polynomially* with $H$, because the algorithms in previous works do not use the variance information. In practice, $H$ is often large, and even a polynomial dependency on $H$ may not be acceptable. Therefore, a natural question is

> **Can we design an algorithm that exploits the variance information to obtain an $\widetilde{O}(\mathrm{poly}(d, \log H)\sqrt{K})$ regret bound for linear mixture MDP?**

### 1.1 Our Contributions

In this paper, we develop new, *variance-aware* confidence sets for linear bandits and linear mixture MDP and answer the above two questions affirmatively.

**Linear Bandits.** For linear bandits, we obtain an $\widetilde{O}(\mathrm{poly}(d)\sqrt{1 + \sum_{k=1}^{K} \sigma_k^2})$ regret bound, where $\sigma_k^2$ is the *unknown* variance at the $k$-th round. To our knowledge, this is the first bound that solely depends on the variance and the feature dimension, and has no explicit polynomial dependency on $K$. When the variance is very small so that $\sigma_k^2 \ll 1$, this bound is substantially smaller than the worst-case $\widetilde{\Theta}(d\sqrt{K})$ bound. Furthermore, this regret bound naturally interpolates between the worst-case $\sqrt{K}$-type bound and the noiseless-case constant-type bound.

**Linear Mixture MDP.** For linear mixture MDP, we obtain the desired $\widetilde{O}(\mathrm{poly}(d, \log H)\sqrt{K})$ regret bound. This is the first regret bound in RL with function approximation that 1) does not scale with the size of the state-action space, and 2) only scales *logarithmically* with the planning horizon

---

[3]$\widetilde{O}(\cdot)$ hides logarithmic factors. Sometimes we write out $\mathrm{polylog}\, H$ explicitly to emphasize the logarithmic dependency on $H$.

[4]This bound holds for setting where the transition is homogeneous and the total reward is bounded by 1. We focus on this setting in this paper. See Section 2 and 3 for more discussions.

$H$. Therefore, we exponentially improve existing results on RL with linear function approximation in term of the $H$ dependency, and resolve an open problem in [Zhou et al., 2020a]. More importantly, our result conveys the positive conceptual message for RL: it is possible to simultaneously overcome the two central challenges in RL, *large state-action space* and *long planning horizon*.

## 1.2 Main Difficulties and Technical Innovations

We first describe limitations of existing works why they cannot achieve the desired regret bounds described above.

**Limitations of Existing Variance-Aware Confidence Sets** Faury et al. [2020], Zhou et al. [2020a] applied Bernstein-style inequalities to construct a confidence sets of the least square estimator for linear bandits. However, their methods can not be applied directly to obtain the desired data-dependent regret bound. Abeille et al. [2021] also designed an variance-dependent confidence set for logistic bandits. However in their problem the rewards are Bernoulli and the variance is a function of the mean.

We give a simple example to illustrate their limitations. Consider the case where the variance is always $\sigma^2 \ll 1$. Let $(\boldsymbol{x}_1, y_1), \ldots, (\boldsymbol{x}_{k-1}, y_{k-1})$ be the samples collected before the $k$-th round. Their confidence set at the $k$-th round is $\Theta_k = \{\boldsymbol{\theta} | \|\boldsymbol{\theta} - \hat{\boldsymbol{\theta}}_k\|_{\Lambda_{k-1}} \leqslant C(\sigma\sqrt{d} + 1 + \lambda^{1/2})\}$ (See In Equation (4.3) of Zhou et al. [2020a] and Theorem 1 of Faury et al. [2020]). where $\Lambda_{k-1} = \sum_{\tau=1}^{k-1} \boldsymbol{x}_\tau \boldsymbol{x}_\tau^\top + \lambda I$ is the un-normalized covariance matrix , $\hat{\boldsymbol{\theta}}_k = \Lambda_{k-1}^{-1} \sum_{\tau=1}^{k-1} y_\tau \boldsymbol{x}_\tau$ is the estimated linear coefficients by least squares, $\lambda$ is a regularization parameter and $C$ is a constant. Consider the case $d = 1$ and $\boldsymbol{x}_k = \sqrt{1/K}$ for $k = 1, \ldots, K$. Their regret bound is roughly

$$\sum_{k=1}^{K} (\sigma\sqrt{d} + 1 + \lambda^{1/2}) \|\boldsymbol{x}_k\|_{\Lambda_k^{-1}} \geqslant (1 + \lambda^{1/2}) \sum_{i=1}^{K} \|\boldsymbol{x}_k\|_{\Lambda_k^{-1}} \geqslant (1 + \lambda^{1/2})\sqrt{\frac{K}{1+\lambda}} \geqslant \sqrt{K},$$

which is much larger than our bound, $O\left(\sqrt{K\sigma^2 + 1}\right)$ when $\sigma$ is very small. For more detailed discussion, please refer to Appendix B.

Below we describe our main techniques.

**Elimination with Peeling.** Instead of using least squares and upper-confidence-bound (UCB), we use an elimination approach. More precisely, for the underlying linear coefficients $\boldsymbol{\theta}^* \in \mathbb{R}^d$, we build a confidence interval for $(\boldsymbol{\theta}^*)^\top \boldsymbol{\mu}$ for every $\boldsymbol{\mu}$ in an $\epsilon$-net of the $d$-dimensional unit ball, and we eliminate $\boldsymbol{\theta} \in \mathbb{R}^d$ if $\boldsymbol{\theta}^\top \boldsymbol{\mu}$ fails to fall in the confidence interval of $(\boldsymbol{\theta}^*)^\top \boldsymbol{\mu}$ for some $\boldsymbol{\mu}$. To build the confidence intervals, we use 1) an empirical Bernstein inequality (cf. Theorem 4) and 2) the peeling technique to both the input norm and the variance magnitude. As will be clear in the proof (cf. Section D), this peeling step is crucial to obtain a tight regret bound for the example above. The new confidence region provides a tighter estimation for $\boldsymbol{\theta}^*$, which helps address the drawback in least squares.

**Generalization of the Elliptical Potential Lemma.** Since we use the peeling technique which comes with a clipping operation, we cannot use the seminal elliptic potential lemma Dani et al. [2008] any more. Instead, we propose a more general lemma below, which provides a bound of potential for a general class of convex functions though with a worse dependency on $d$ than the bound in the elliptical potential lemma. We believe this lemma can be applied to other problems as well.

**Lemma 1** (Generalized Quadratic Potential Lemma). *Let $f(x) \geqslant 0$ be a convex function over $\mathbb{R}$ such that $\frac{f(x)}{x^2} \leqslant \frac{f(y)}{y^2} \leqslant 1$ and $f(x) \geqslant f(y)$ if $x^2 \geqslant y^2 > 0$. Let $\mathbb{B}(1)$ denote the $d$-dimensional unit ball. Fix $\ell \in (0, 1]$. For any $\boldsymbol{x}_1, \boldsymbol{x}_2, \ldots, \boldsymbol{x}_t \in \mathbb{B}(1)$ and $\boldsymbol{\mu}_1, \boldsymbol{\mu}_2, \ldots, \boldsymbol{\mu}_t \in \mathbb{B}(1)$, we have that*

$$\sum_{i=1}^{t} \min \left\{ \frac{f(\boldsymbol{x}_i \boldsymbol{\mu}_i)}{\sum_{j=1}^{i-1} f(\boldsymbol{x}_j \boldsymbol{\mu}_i) + \ell^2}, 1 \right\} \leqslant O(d^4 \log(dt/\ell)).$$

Note that by choosing $f(x) = x^2$ and $\boldsymbol{\mu}_i = \frac{\boldsymbol{x}_i \Lambda_i^{-1}}{\|\boldsymbol{x}_i \Lambda_i^{-1}\|}$ with $\Lambda_i = \sum_{j=1}^{i-1} \boldsymbol{x}_j \boldsymbol{x}_j^\top + \ell\mathbf{I}$, Lemma 1 reduces to the classical elliptic potential lemma [Dani et al., 2008]. Our proof consists of two major parts.

We first establish a symmetric version of Equation (**??**) using rearrangement inequality, and then bound the number of times the energy for some $\boldsymbol{\mu}$ (i.e., $\sum_{j=1}^{i} f(\boldsymbol{x}_j \boldsymbol{\mu}) + l^2$) doubles. The full proof is deferred to Appendix C.

For linear mixture MDP, we propose another technique to further reduce the dependency on $d$.

**Recursion-based Variance Estimation.**    In linear bandits, generally it is not possible to estimate the variance because the variance at each round can arbitrarily different. On the other hand, for linear mixture MDP, the variance is a quadratic function of the underlying coefficient $\boldsymbol{\theta}^*$. Furthermore, the higher moments are polynomial functions of $\boldsymbol{\theta}^*$. Utilizing this rich structure and leveraging the recursion idea in previous analyses on tabular RL [Lattimore and Hutter, 2012, Li et al., 2020, Zhang et al., 2020a], we explicitly estimate the variance and higher moments to further reduce the regret. See Section 5 for more explanations.

## 2   Related Work

**Linear Bandits.**    There is a line of theoretical analyses of linear bandits problems [Auer et al., 2002, Dani et al., 2008, Chu et al., 2011, Abbasi-Yadkori et al., 2011, Li et al., 2019a,b]. For infinite-actioned linear bandits, the minimax regret bound is $\widetilde{\Theta}(d\sqrt{K})$. and recent works tried to give fine-grained instance-dependent bounds [Katz-Samuels et al., 2020, Jedra and Proutiere, 2020]. For multi-armed bandits, Audibert et al. [2006] showed by exploiting the variance information, one can improve the regret bound. For linear bandits, only a few work studied how to use the variance information. Faury et al. [2020] studied logistic bandit problem with adaptivity to the variance of noise, where a Bernstein-style confidence set was proposed. However, they assume the variance is known and cannot attain the desired variance-dependent bound due to the example we gave above. Linear bandits can be also seen as a simplified version of RL with linear function approximation, where the planning horizon degenerates to $H = 1$.

**RL with Linear Function Approximation.**    Recently, it is a central topic in the theoretical RL community to figure out the necessary and sufficient conditions that permit efficient learning in RL with large state-action space [Wen and Van Roy, 2013, Jiang et al., 2017, Yang and Wang, 2019, 2020, Du et al., 2019b, 2020a, 2019a, 2020b, Jiang et al., 2017, Feng et al., 2020, Sun et al., 2019, Dann et al., 2018, Krishnamurthy et al., 2016, Misra et al., 2019, Ayoub et al., 2020, Zanette et al., 2020, Wang et al., 2019, 2020c,b, Jin et al., 2020, Weisz et al., 2020, Modi et al., 2020, Shariff and Szepesvári, 2020, Jin et al., 2020, Cai et al., 2019, He et al., 2020, Zhou et al., 2020a]. However, to our knowledge, all existing regret upper bounds have a polynomial dependency on the planning horizon $H$, except works that assume the environment is deterministic [Wen and Van Roy, 2013, Du et al., 2020b].

This paper studies the linear mixture MDP setting [Ayoub et al., 2020, Zhou et al., 2020b,a, Modi et al., 2020], which assumes the underlying transition is a linear combination of some known base models. Ayoub et al. [2020] gave an algorithm, UCRL-VTR, with an $\widetilde{O}(dH^2\sqrt{K})$ regret in the time-inhomogeneous model.[5] Our algorithm improves the $H$-dependency from $\mathrm{poly}(H)$ to $\mathrm{polylog}(H)$, at the cost of a worse dependency on $d$.

**Variance Information in Tabular MDP.**    The use of the variance information in tabular MDP was first proposed by Lattimore and Hutter [2012] in the discounted MDP setting, and was later adopted in the episodic MDP setting [Azar et al., 2017, Jin et al., 2018, Zanette and Brunskill, 2019, Dann et al., 2019, Zhang et al., 2020a,b]. This technique is crucial to tighten the dependency on $H$.

**Concurrent Work by Zhou et al. [2020a].**    While preparing this draft, we noticed a concurrent work by Zhou et al. [2020a], who also studied how to use the variance information for linear

---

[5]The time-inhomogeneous model refers to the setting where the transition probability can vary at different levels, and the time-homogeneous model refers to the setting where the transition probability is the same at different levels. Roughly speaking, the model complexity of the time-inhomogeneous model is $H$ times larger than that of the time-homogeneous model. In general, it is straightforward to tightly extend a result for the time-homogeneous model to the time-inhomogeneous model by extending the state-action space [Jin et al., 2018, Footnote 2], but not vice versa.

bandits and linear mixture MDPs. We first compare their results with ours. For linear bandits, they proved an $\widetilde{O}(\sqrt{dK} + d\sqrt{\sum_{i=1}^{K} \sigma_i^2})$ regret bound, while we prove an $\widetilde{O}(d^{4.5}\sqrt{\sum_{i=1}^{K} \sigma_i^2} + d^5)$ regret bound. Our bound has a worse dependency on $d$, but in the regime where $K$ is very large and the sum of the variances is small, our bound is stronger. Furthermore, they assumed *the variance is known while we do not need this assumption*. For linear mixture MDP, they proved an $\widetilde{O}(\sqrt{d^2H + dH^2}\sqrt{K} + d^2H^2 + d^3H)$ bound for the time-inhomogeneous model, while we prove an $\widetilde{O}(d^{4.5}\sqrt{K} + d^5) \times \text{polylog}(H)$ bound for the time-homogeneous model. Their bound has a better dependency on $d$ than ours and is near-optimal in the regime $K = \Omega(\text{poly}(d, H))$ and $H = O(d)$. On the other hand, we have an exponentially better dependency on $H$ in the time-homogeneous model. Indeed, obtaining a regret bound that is logarithmic in $H$ (in the time-homogeneous model) was raised as an open question in their paper [Zhou et al., 2020a, Remark 5.5].

Next, we compare the algorithms and the analyses. The algorithms in the two papers are very different in nature: ours are based on elimination while theirs are based on least squares and UCB. We note that, for linear bandits, their current analysis cannot give a $\sqrt{K}$-free bound because there is a term that scales *inversely* with the variance. This can be seen by plugging the first line of their (B.25) to their (B.23). For the same reason, they cannot give a horizon-free bound in the time-homogeneous linear mixture MDP. In sharp contrast, our analysis does not have the term depending on the inverse of the variance. On the other hand, their algorithms are computationally efficient (given certain computation oracles), but our algorithms are not because ours are elimination-based. See Section 6 for more discussions.

## 3 Preliminaries

**Notations.** We use $\mathbb{B}_p^d(r) = \{x \in \mathbb{R}^d : \|x\|_p \leqslant r\}$ to denote the $d$-dimensional $\ell_p$-ball of radius $r$, so $\mathbb{B}(1) = \mathbb{B}_2^d(1)$ For any set $S \subseteq \mathbb{R}^d$, we use $\partial S$ to denote its boundary. For $N \in \mathbb{N}$, we define $[N] = \{1, \ldots, N\}$. One important operation used in our algorithms and analyses is clipping. Given $\ell > 0$ and $u \in \mathbb{R}$, we define

$$\text{clip}(u, \ell) = \min\{|u|, \ell\} \cdot \frac{u}{|u|}$$

for $u \neq 0$ and $\text{clip}(0, \ell) = 0$. For any two vectors $\boldsymbol{u}, \boldsymbol{v}$, to save notations, we use $\boldsymbol{u}\boldsymbol{v} = \boldsymbol{u}^\top \boldsymbol{v}$ to denote their inner product when no ambiguity.

**Linear Bandits.** We use $K$ to denote the number of rounds in the linear bandits. At each round $k = 1, \ldots, K$, the algorithm is first given the context set $\mathcal{A}_k \subseteq \mathbb{B}_2^d(1)$, then the algorithm chooses an action $\boldsymbol{x}_k \in \mathcal{A}_k$ and receives the noisy reward $r_k = \boldsymbol{x}_k\boldsymbol{\theta}^* + \varepsilon_k$, where $\boldsymbol{\theta}^* \in \mathbb{B}_2^d(1)$ is the unknown underlying linear coefficients and $\varepsilon_k$ is the random noise. We define $\mathcal{F}_k = \sigma(\boldsymbol{x}_1, \varepsilon_1, \ldots, \boldsymbol{x}_k, \varepsilon_k, \boldsymbol{x}_{k+1})$. We assume that $|r_k| \leqslant 1$ and that the noise $\varepsilon_k$ satisfies $\mathbb{E}[\varepsilon_k \mid \mathcal{F}_k] = 0$ and $\mathbb{E}[\varepsilon_k^2 \mid \mathcal{F}_k] = \sigma_k^2$. The goal is to learn $\boldsymbol{\theta}^*$ and minimize the cumulative expected regret $\mathbb{E}[\mathfrak{R}^K]$, where

$$\mathfrak{R}^K = \sum_{k=1}^{K} [\max_{\boldsymbol{x} \in \mathcal{A}_k} \boldsymbol{x}\boldsymbol{\theta}^* - \boldsymbol{x}_k\boldsymbol{\theta}^*].$$

**Remark 1.** *Here we assume the reward is uniformly bounded ($|r_k| \leqslant 1$) instead of 1-sub-Gaussian commonly used in the literature only for the ease of presentation, because in RL, it is standard to assume bounded reward. Note if the noise is 1-sub-Gaussian, our algorithm also applies with only an $O(\log K)$ overhead because a problem with 1-sub-Gaussian noise can be reduced to that with uniformly bounded noise by clipping the noise with a threshold $O(\log K)$.*

**Episodic MDP and Linear Mixture MDP.** We use a tuple $(\mathcal{S}, \mathcal{A}, r, P, K, H)$ to define an episodic finite-horizon MDP. Here, $\mathcal{S}$ is its state space, $\mathcal{A}$ is its action space, $r : \mathcal{S} \times \mathcal{A} \to [0, 1]$ is its reward function, $P(s' \mid s, a)$ is the transition probability from the state-action pair $(s, a)$ to the new state $s'$, $K$ is the number of episodes, and $H$ is the planning horizon of each episode. Without the loss of generality, we assume a fixed initial state $s_1$. A sequence of functions $\pi = \{\pi_h : \mathcal{S} \to \triangle(\mathcal{A})\}_{h=1}^{H}$ is an policy, where $\triangle(\mathcal{A})$ denotes the set of all possible distributions over $\mathcal{A}$.

At each episode $k = 1, \ldots, K$, the algorithm outputs a policy $\pi^k$, which is then executed on the MDP by $a_h^k \sim \pi_h^k(s_h^k), s_{h+1}^k \sim P(\cdot \mid s_h^k, a_h^k)$. We let $r_h^k = r(s_h^k, a_h^k)$ be the reward at time step $h$ in

**Algorithm 1** VOFUL: **V**ariance-Aware **O**ptimism in the **F**ace of **U**ncertainty for **L**inear Bandits

---

1: **Initialize:** $\ell_i = 2^{2-i}, \iota = 16d\ln\frac{dK}{\delta}, L_2 = \lceil\log_2 K\rceil, \Lambda_2 = \{1, 2, \ldots, L_2 + 1\}, \Theta_1 = \mathbb{B}_2^d(1)$,
   Let $\mathcal{B}$ be an $K^{-3}$-net of $\mathbb{B}_2^d(2)$ with size not larger than $(\frac{4}{K})^{3d}$
2: **for** $k = 1, 2, \ldots, K$ **do**
3:     **Optimistic Action Selection:**
4:     Observe context set $\mathcal{A}_k \subseteq \mathbb{B}_2^d(1)$
5:     Compute $\boldsymbol{x}_k \leftarrow \arg\max_{\boldsymbol{x}\in\mathcal{A}_k} \max_{\boldsymbol{\theta}\in\Theta_k} \boldsymbol{x}\boldsymbol{\theta}$, choose action $\boldsymbol{x}_k$
6:     Receive feedback $y_k$
7:     **Construct Confidence Set:**
8:     For each $\boldsymbol{\theta}\in\mathbb{B}_2^d(1)$, define $\epsilon_k(\boldsymbol{\theta}) = y_k - \boldsymbol{x}_k\boldsymbol{\theta}, \eta_k(\boldsymbol{\theta}) = (\epsilon_k(\boldsymbol{\theta}))^2$.
9:     Define confidence set $\Theta_{k+1} = \bigcap_{j\in\Lambda_2}\Theta_{k+1}^j$, where

$$\Theta_{k+1}^j = \left\{\boldsymbol{\theta}\in\mathbb{B}_2^d(1) : \left|\sum_{v=1}^k \mathsf{clip}_j(\boldsymbol{x}_v\boldsymbol{\mu})\epsilon_v(\boldsymbol{\theta})\right| \leqslant \sqrt{\sum_{v=1}^k \mathsf{clip}_j^2(\boldsymbol{x}_v\boldsymbol{\mu})\eta_v(\boldsymbol{\theta})\iota + \ell_j\iota}, \forall\boldsymbol{\mu}\in\mathcal{B}\right\} \quad (1)$$

    and $\mathsf{clip}_j(\cdot) = \mathsf{clip}(\cdot, \ell_j)$.
10: **end for**

---

episode $k$. Importantly, we assume the transition model $P(\cdot \mid \cdot, \cdot)$ is time-homogeneous, which is necessary to bypass the $\mathrm{poly}(H)$ dependency. We assume that the reward function is known, which is standard in the theoretical RL literature to simplify the presentation [Modi et al., 2020, Ayoub et al., 2020]. We let $\pi^*$ to denote the optimal policy which achieves the maximum reward in expectation.

We make the following regularity assumption on the rewards: the sum of reward, $\sum_{h=1}^H r_h$, in each episode is bounded by 1.

**Assumption 2** (Non-uniform reward). $\sum_{h=1}^H r_h^k \leqslant 1$ *almost surely for any policy $\pi^k$.*

This assumption is much weaker than the common assumption where the reward at each time step is bounded by $1/H$ (uniform reward) because Assumption 2 allows one spiky reward as large as $\Omega(1)$. See more discussions about this reward scaling in Jiang and Agarwal [2018], Wang et al. [2020a], Zhang et al. [2020a].

For any policy $\pi$, we define its $H$-step $V$-function and $Q$-function as

$$V_h^\pi(s) = \max_{a\in\mathcal{A}} Q_h^\pi(s, a)$$
$$\text{where } Q_h^\pi(s, a) = r(s, a) + \mathbb{E}_{s'\sim P(\cdot|s,a)} V_{h+1}^\pi(s') \text{ for } h = 1, \ldots, H$$

where we set $V_{H+1} = 0$. For simplicity, we also denote $V^\pi(s_1) = V_1^\pi(s_1)$ and $V^*(s_1) = V^{\pi^*}(s_1)$.

A linear mixture MDP is an episodic MDP with the extra assumption that its transition model is an unknown linear combination of a known set of models. Specifically, there is an unknown parameter $\boldsymbol{\theta}^* \in \mathbb{B}_1^d(1)$, such that $P = \sum_{i=1}^d \theta_i^* P_i$ where based models $P_1, \ldots, P_d$ are given. The goal is to learn $\boldsymbol{\theta}^*$ and minimize the cumulative expected regret $\mathbb{E}[\mathfrak{R}^K]$, where

$$\mathfrak{R}^K = \sum_{k=1}^k [V^*(s_1) - V^k(s_1)].$$

# 4 Algorithm and Theory for Linear Bandits

In this section, we introduce our algorithm for linear bandits and analyze its regret. The pseudo-code is listed in Algorithm 1. The following theorem shows our algorithm achieves the desired variance-dependent regret bound. The full proof is deferred to Section D.

**Theorem 3.** *The expected regret of Algorithm 1 is bounded by $\mathbb{E}[\mathfrak{R}^K] \leqslant \tilde{O}(d^{4.5}\sqrt{\sum_{k=1}^K \sigma_k^2} + d^5)$.*

This theorem shows our algorithm's regret has no explicit polynomial dependency on the number of rounds $K$. In the worst-case where the variance is $\Omega(1)$, our bound becomes $\tilde{O}(d^{4.5}\sqrt{K} + d^5)$, which has a worse dependency on $d$ compared with the minimax optimal algorithms [Dani et al.,

2008, Abbasi-Yadkori et al., 2011]. However, in the benign case where the variance is $o(1)$, our bound can be much smaller. In particular, in the noiseless case, our bound is a constant-type regret bound, up to logarithmic factors. One future direction is to design an algorithm that is minimax optimal in the worst-case but also adapts to the variance magnitude like ours.

## 4.1 Main Algorithm

Now we describe our algorithm. Similar to many other linear bandit algorithms, the algorithm maintains confidence sets $\{\Theta_k\}_{k \geq 1}$ for the underlying parameter $\theta^*$, and then choose the action greedily according to the confidence set.

To relax the known variance assumption, we use the following empirical Bernstein inequality that depends on the *empirical variance*, in contrast to the Bernstein inequality that depends on the *true variance*, which was used in existing works [Zhou et al., 2020b, Faury et al., 2020].

**Theorem 4.** *Let $\{\mathcal{F}_i\}_{i=0}^n$ be a filtration. Let $\{X_i\}_{i=1}^n$ be a sequence of real-valued random variables such that $X_i$ is $\mathcal{F}_i$-measurable. We assume that $\mathbb{E}[X_i \mid \mathcal{F}_{i-1}] = 0$ and that $|X_i| \leq b$ almost surely. For $\delta < e^{-1}$, we have*

$$\Pr\left[ \left| \sum_{i=1}^n X_i \right| \leq 8\sqrt{\sum_{i=1}^n X_i^2 \ln \frac{1}{\delta}} + 16b \ln \frac{1}{\delta} \right] \geq 1 - 6\delta \log_2 n. \tag{2}$$

Importantly, this inequality controls the deviation via the empirical variance, which is $X_i^2$ and can be computed once $X_i$ is known. Note some previously proved inequalities require certain independence assumptions and thus cannot be directly applied to martingales [Maurer and Pontil, 2009, Peel et al., 2013], so they cannot be used for solving our linear bandits problem. The proof of the theorem is deferred to Appendix D.2.

More effort is devoted to designing a confidence set that fully exploits the variance information. Note Theorem 4 is for real-valued random variables, and it remains unclear how to generalize it to the linear regression setting, which is crucial for building confidence sets for linear bandits. Previous works built up their confidence sets based on analyzing the ordinary ridged least square estimator [Dani et al., 2008, Abbasi-Yadkori et al., 2011], or the weighted one [Zhou et al., 2020a].

We drop the least square estimators and instead, we take a testing-based approach, as done in Equation (1). To illustrate the idea, we first ignore the $\mathrm{clip}_j(\cdot)$ operation and $\ell_j$ terms. We define the noise function $\epsilon_k(\theta)$ and the variance function $\eta_k(\theta)$ (Line 8 of Algorithm 1). Note that $\epsilon_k(\theta^*) = \varepsilon_k$ and $\eta_k(\theta^*) = \varepsilon_k^2$, so we have the following fact: if $\theta = \theta^*$, then Equation (2) would be true if we replace $X_k = w_k(\mu)\epsilon_k(\theta)$ and $X_k^2 = w_k^2(\mu)\eta_k(\theta)$ with high probability, where $\{w_k(\mu)\}$ is a proper sequence of weights depending on the test direction $\mu$. Our approach uses the fact in the opposite direction: if weighted $w_k(\mu)\epsilon_k(\theta), w_k^2(\mu)\eta_k(\theta)$ satisfies Equation (2) for all possible test directions $\mu$ in an $K^{-3}$-net of the $d$-dimensional unit ball, then we put $\theta$ into the confidence set.

**Remark 2.** *One can also view the algorithm as an elimination-based algorithm: if there exists some test direction $\mu$ such that Equation (2) fails for $X_k = w_k(\mu)\epsilon_k(\theta)$ and $X_k^2 = w_k^2(\mu)\eta_k(\theta)$, then we eliminate $\theta$ from the confidence set permanently.*

Given the test direction $\mu$, following the least square estimation, $w_k(\mu)$ is set to be $x_k\mu$. However, with $w_k(\mu) = x_k\mu$, the right-hand-side of Equation (2) is at least $b \geq \max_{1 \leq k \leq n} |w_k(\mu)| = \max_{1 \leq k \leq n} |x_k\mu|$, which might be dominant compared with $\sum_{k=1}^n w_k^2(\mu)\eta_k(\theta)$ (See Appendix B for a toy example). To address this problem, we consider to peel $w_k(\mu)$ for various thresholds of difference level. More precisely, we construct confidence regions respectively with $w_k^j(\mu) = \mathrm{clip}_j(x_k\mu)$, where $l_j = 2^{2-j}$ for $j = 1, 2, \ldots, \lceil \log_2 K \rceil$. At last, we define the final confidence region as the intersections of all these confidence regions.

**Remark 3.** *Note that existing confidence sets in Equation (1) either do not exploit variance information [Dani et al., 2008, Abbasi-Yadkori et al., 2011], or require the variance to be known and do not fully exploit the variance information [Zhou et al., 2020a, Faury et al., 2020] as their regret bounds still have an $\widetilde{O}(\sqrt{K})$ term.*

## 4.2 Proof Sketch of Theorem 3

Now we explain how our confidence set enables us to obtain a variance-dependent regret bound. We define $\boldsymbol{\theta}_k = \arg\max_{\boldsymbol{\theta} \in \Theta_k} \boldsymbol{x}_k(\boldsymbol{\theta} - \boldsymbol{\theta}^*)$ and $\boldsymbol{\mu}_k = \boldsymbol{\theta}_k - \boldsymbol{\theta}^*$. Then our goal is to bound the regret $\sum_k \boldsymbol{x}_k \boldsymbol{\mu}_k$. Our main idea is to consider $\{\boldsymbol{x}_k\}, \{\boldsymbol{\mu}_k\}$ as two sequences of vectors. We decouple the complicated dependency between $\{\boldsymbol{x}_k\}$ and $\{\boldsymbol{\mu}_k\}$ by a union bound over the net $\mathcal{B}$ (defined in Line 1 of Algorithm 1). To bound the regret, we implicitly divide all rounds $k \in [K]$ into norm layers based on $\log_2 |\boldsymbol{x}_k \boldsymbol{\mu}_k|$ in the analysis. [6] Within each layer, we apply Equation (1) to obtain the relations between $\boldsymbol{\mu}_k$ and $\{\boldsymbol{x}_1, \ldots, \boldsymbol{x}_{k-1}\}$, which would self-normalize the growth of the two sequences, ensuring that their in-layer total sum is properly bounded. Since we have logarithmically many layers, the total regret is then properly bounded. We highlight that our norm peeling technique ensures that the variance-dependent term dominates the other variance-independent term in Bernstein inequalities ($\sqrt{\sum X_i^2} \gtrsim b$ in Theorem 4), which resolves the variance-independent term in the final regret bound obtained by Zhou et al. [2020a].

We start the analysis by proving that the probability of failure events (i.e., the events where $\boldsymbol{\theta}^* \notin \Theta_k$ for some $k \in [K]$) is properly bounded (see Lemma 18). Assuming the successful events happen, we have that $\boldsymbol{\theta}^* \in \Theta_k$ for all $k \in [K]$. Then we obtain that.

$$\mathfrak{R}^K := \sum_{k=1}^K \left( \max_{\boldsymbol{x} \in \mathcal{A}_k} \boldsymbol{\theta}^* - \boldsymbol{x}_k \boldsymbol{\theta}^* \right) \leqslant \sum_{k=1}^K \max_{\boldsymbol{x} \in \mathcal{A}_k, \boldsymbol{\theta} \in \Theta_k} \boldsymbol{x}_k (\boldsymbol{\theta}_k - \boldsymbol{\theta}^*) \leqslant \sum_{k=1}^K \boldsymbol{x}_k (\boldsymbol{\theta}_k - \boldsymbol{\theta}^*) = \sum_{k=1}^K \boldsymbol{x}_k \boldsymbol{\mu}_k.$$

Next we divide the time steps $[K]$ into $L_2 + 1$ disjoint subsets $\{\mathcal{K}_j\}_{j=1}^{L_2+1}$ according to the magnitude of $\boldsymbol{x}_k \boldsymbol{\mu}_k$. More precisely, for for $1 \leqslant j \leqslant L_2$ we assign $k$ to $\mathcal{K}_j$ iff $\boldsymbol{x}_k \boldsymbol{\mu}_k \in (l_j/2, l_j]$, and for $j = L_2 + 1$, we assign $k$ to $\mathcal{K}_j$ iff $\boldsymbol{x}_k \boldsymbol{\mu}_k \leqslant l_{l_2+1}/2$. Define

$$\Phi_k^j(\boldsymbol{\mu}) = \sum_{v=1}^{k-1} \mathsf{clip}_j(\boldsymbol{x}_v \boldsymbol{\mu}) \boldsymbol{x}_v \boldsymbol{\mu} + \ell_j^2, \qquad \Psi_k^j(\boldsymbol{\mu}) = \sum_{v=1}^{k-1} \mathsf{clip}_j^2(\boldsymbol{x}_v \boldsymbol{\mu}) \eta_v(\boldsymbol{\theta}^*). \tag{3}$$

By the definition of $\Theta_k$ in (1), we have that (see Claim 20)

$$\sum_{k=1}^K \boldsymbol{x}_k \boldsymbol{\mu}_k \leqslant 1 + \sum_{j=1}^{L_2} \sum_{k \in \mathcal{K}_j} \boldsymbol{x}_k \boldsymbol{\mu}_k \times \frac{3\sqrt{\Psi_k^j(\boldsymbol{\mu}_k)\iota} + \sqrt{\sum_{v=1}^{k-1} 2\mathsf{clip}_j^2(\boldsymbol{x}_v \boldsymbol{\mu}_k)(\boldsymbol{x}_v \boldsymbol{\mu}_k)^2 \iota} + 3\ell_j \iota}{\Phi_k^j(\boldsymbol{\mu}_k)}. \tag{4}$$

Continuing the computation, we have that

$$\sum_{j=1}^{L_2} \sum_{k \in \mathcal{K}_j} \boldsymbol{x}_k \boldsymbol{\mu}_k \frac{\sqrt{\sum_{v=1}^{k-1} 2\mathsf{clip}_j^2(\boldsymbol{x}_v \boldsymbol{\mu}_k)(\boldsymbol{x}_v \boldsymbol{\mu}_k)^2 \iota}}{\Phi_k^j(\boldsymbol{\mu}_k)}$$

$$\leqslant \frac{1}{2} \sum_{j=1}^{L_2} \sum_{k \in \mathcal{K}_j} \boldsymbol{x}_k \boldsymbol{\mu}_k + \sum_{j=1}^{L_2} \sum_{k \in \mathcal{K}_j} \boldsymbol{x}_k \boldsymbol{\mu}_k \mathbb{I} \left\{ \frac{\sqrt{\sum_{v=1}^{k-1} 2\mathsf{clip}_j^2(\boldsymbol{x}_v \boldsymbol{\mu}_k)(\boldsymbol{x}_v \boldsymbol{\mu}_k)^2 \iota}}{\Phi_k^j(\boldsymbol{\mu}_k)} > \frac{1}{2} \right\}$$

$$\leqslant \frac{1}{2} \sum_{j=1}^{L_2} \sum_{k \in \mathcal{K}_j} \boldsymbol{x}_k \boldsymbol{\mu}_k + \sum_{j=1}^{L_2} \sum_{k \in \mathcal{K}_j} \boldsymbol{x}_k \boldsymbol{\mu}_k \frac{4l_j \iota}{\Phi_k^j(\boldsymbol{\mu}_k)} \tag{5}$$

$$\leqslant \frac{1}{2} \sum_{j=1}^{L_2} \sum_{k \in \mathcal{K}_j} \boldsymbol{x}_k \boldsymbol{\mu}_k + O(d^4 |\Lambda_2| \iota \log^3(dK)), \tag{6}$$

---

[6]This cannot be done explicitly in the algorithm, since it would re-couple the two sequences.

---
**Algorithm 2** VARLin: **V**ariance-**A**ware **RL** with **Lin**ear Function Approximation
---
1: **Initialize:** $\ell_i = 2^{2-i}$, $\iota = 16d \ln \frac{dHK}{\delta}$, $L_0 = \lceil \log_2 KH \rceil$, $L_1 = L_2 = \lceil 5 \log_2(HK) + 3 \rceil$, $\Lambda_0 = \{0, 1, , \ldots, L_0\}$, $\Lambda_1 = \{1, \ldots, L_1\}$, $\Lambda_2 = \{1, \ldots, L_2\}$. $\mathcal{B}$ be an $(HK)^{-3}$-net of $\mathbb{B}_1^d(2)$ with size no larger than $(\frac{4}{HK})^{3d}$. $\Theta_1 = \mathbb{B}_1^d(1)$.

2: **for** $k = 1, 2, \ldots, K$ **do**

3:    **Optimistic Planning**:

4:    **for** $h = H, H-1, \ldots, 1$ **do**

5:        For each $(s, a) \in \mathcal{S} \times \mathcal{A}$, let $Q_h^k(s, a) = \min\{1, r(s, a) + \max_{\boldsymbol{\theta} \in \Theta_k} \sum_{i=1}^d \theta_i P_{s,a}^i V_{h+1}^k\}$.

6:        For each $s \in \mathcal{S}$, let $V_h^k(s) = \max_{a \in \mathcal{A}} Q_h^k(s, a)$.

7:    **end for**

8:    **for** $h = 1, 2, \ldots, H$ **do**

9:        Choose action $a_h^k \leftarrow \arg\max_{a \in \mathcal{A}} Q_h^k(s_h^k, a)$, observe the next state $s_{h+1}^k$.

10:   **end for**

11:   **Construct Confidence Set**:

12:   For $m \in \Lambda_0, h \in [H]$, define the input $\boldsymbol{x}_{k,h}^m = [P_{s_h^k, a_h^k}^1 (V_{h+1}^k)^{2^m}, \ldots, P_{s_h^k, a_h^k}^d (V_{h+1}^k)^{2^m}]^\top$.

13:   For $m \in \Lambda_0, h \in [H]$, define the variance estimate $\eta_{k,h}^m = \max_{\boldsymbol{\theta} \in \Theta_k} \{\boldsymbol{\theta} \boldsymbol{x}_{k,h}^{m+1} - (\boldsymbol{\theta} \boldsymbol{x}_{k,h}^m)^2\}$.

14:   Denote $\epsilon_{v,u}^m(\boldsymbol{\theta}) = \boldsymbol{\theta} \boldsymbol{x}_{v,u}^m - (V_{u+1}^v(s_{u+1}^v))^{2^m}$ for $m \in \Lambda_0, u \in [H], v \in [k-1]$

15:   Define $\mathcal{T}_{k+1}^{m,i} = \{(v, u) \in [k] \times [H] : \eta_{v,u}^m \in (\ell_{i+1}, \ell_i]\}$, $\mathcal{T}_{k+1}^{m, L_1+1} = \{(v, u) \in [k] \times [H] : \eta_{v,u}^m \leq \ell_{L_1+1}\}$.

16:   Define the confidence ball $\Theta_{k+1} = \bigcap_{m,i,j} \Theta_{k+1}^{m,i,j}$, where

$$\Theta_{k+1}^{m,i,j} = \left\{ \boldsymbol{\theta} \in \mathbb{B}_1^d(1) : \left| \sum_{(v,u) \in \mathcal{T}_k^{m,i}} \mathsf{clip}_j(\boldsymbol{x}_{v,u}^m \boldsymbol{\mu}) \epsilon_{v,u}^m(\boldsymbol{\theta}) \right| \right.$$

$$\left. \leq 4\sqrt{\sum_{(v,u) \in \mathcal{T}_k^{m,i}} \mathsf{clip}_j^2(\boldsymbol{x}_{v,u}^m \boldsymbol{\mu}) \eta_{v,u}^m \iota} + 4\ell_j \iota, \forall \boldsymbol{\mu} \in \mathcal{B} \right\} \qquad (7)$$

and $\mathsf{clip}_j(\cdot) = \mathsf{clip}(\cdot, \ell_j)$

17: **end for**

---

where (5) is by the fact that $\frac{\sqrt{\sum_{v=1}^{k-1} 2\mathsf{clip}_j^2(\boldsymbol{x}_v \boldsymbol{\mu}_k)(\boldsymbol{x}_v \boldsymbol{\mu}_k)^2 \iota}}{\Phi_k^j(\boldsymbol{\mu}_k)} > \frac{1}{2}$ implies that $\frac{4l_j \iota}{\Phi_k^j(\boldsymbol{\mu}_k)} > 1$, and (6) follows by Lemma 17. By (4) and (6), we have that

$$\sum_{k=1}^K \boldsymbol{x}_k \boldsymbol{\mu}_k \leq 12 \sum_{j=1}^{L_2} \sum_{k \in \mathcal{K}_j} \boldsymbol{x}_k \boldsymbol{\mu}_k \times \frac{\sqrt{\Psi_k^j(\boldsymbol{\mu}_k)\iota}}{\Phi_k^j(\boldsymbol{\mu}_k)} + \tilde{O}(d^5)$$

$$\leq \sum_{j=1}^{L_2} \sum_{k \in \mathcal{K}_j} \frac{12 \boldsymbol{x}_k \boldsymbol{\mu}_k \ell_j}{\Phi_k^j(\boldsymbol{\mu}_k)} \sqrt{\sum_{k=1}^K \eta_k(\boldsymbol{\theta}^*)\iota} + \tilde{O}(d^5) \leq O(d^4|\Lambda_2| \log^3(dK))\sqrt{\left(\ln\frac{1}{\delta} + \sum_{k=1}^K \sigma_k^2\right)\iota} + \tilde{O}(d^5),$$

where the last inequality uses Lemma 17. Therefore, the regret bound is $\tilde{O}\left(d^{4.5}\sqrt{\sum_{k=1}^K \sigma_k^2} + d^5\right)$.

See Section D for the full proof.

## 5 Algorithm and Theory for Linear Mixture MDP

We introduce our algorithm and the regret bound for linear mixture MDP. Its pseudo-code is listed in Algorithm 2 and its regret bound is stated below. The proof is deferred to Section E.

**Theorem 5.** *The expected regret of Algorithm 2 is bounded by* $\mathbb{E}[\mathfrak{R}^K] \leq \tilde{O}\left(d^{4.5}\sqrt{K} + d^9\right)$.

Before describing our algorithm, we introduce some additional notations. In this section, we assume that, unless explicitly stated, the variables $m, i, j, k, h$ iterate over the sets $\Lambda_0, \Lambda_1, \Lambda_2, [K], [H]$, respectively. See Line 1 of Algorithm 2 for the definitions of these sets. For example, at Line 16 of Algorithm 2, we have $\bigcap_{m,i,j} \Theta_{k+1}^{m,i,j} = \bigcap_{m \in \Lambda_0, i \in \Lambda_1, j \in \Lambda_2} \Theta_{k+1}^{m,i,j}$.

The starting point of our algorithm design is from Zhang et al. [2020a], in which the authors obtained a nearly horizon-free regret bound in tabular MDP. A natural idea is to combine their proof with our results for linear bandits and obtain a nearly horizon-free regret bound for linear mixture MDP.

Note that, however, there is one caveat for such direct combination: in Section 4, the confidence set $\Theta_k$ is updated at a per-round level, in that $\Theta_k$ is built using all rounds prior to $k$; while for the RL setting, the confidence set $\Theta_k$ could only be updated at a per-episode level and use all time steps prior to episode $k$. Were it updated at a per-time-step level, severe dependency issues would prevent us from bounding the regret properly. Such discrepancy in update frequency results in a gap between the confidence set built using data prior to episode $k$, and that built using data prior to time step $(k, h)$. Fortunately, we are able to resolve this issue. In Lemma 22, we show that we can relate these two confidence intervals, except for $\tilde{O}(d)$ "bad" episodes. Therefore, we could adapt the analysis in Zhang et al. [2020a] only for the not "bad" episodes, and we bound the regret by 1 for the "bad" episodes. The resulting regret bound should be $\tilde{O}(d^{6.5}\sqrt{K})$.

To further reduce the horizon-free regret bound to $\tilde{O}(d^{4.5}\sqrt{K})$, we present another novel technique. We first note an important advantage of the linear mixture MDP setting over the linear bandit setting: in the latter setting, we cannot estimate the variance because there is no structure on the variance among different actions; while in the former setting, we could estimate an upper bound of the variance, because the variance is a quadratic function of $\boldsymbol{\theta}^*$. Therefore, we can use the peeling technique on the *variance magnitude* to reduce the regret (comparing Equation (30) and Equation (43) in appendix). We note that one can also apply this step to linear bandits if the variance can be estimated.

Along the way, we also need to bound the gap between estimated variance and true variance, which can be seen as the "regret of variance predictions." Using the same idea, we can build a confidence set using the variance sequence $(\boldsymbol{x}^2)$, and the regret of variance predictions can be bounded by the variance of variance, namely the 4-th moment. Still, a peeling step on the 4-th moment is required to bound the regret of variance predictions, we need to bound the gap between estimated 4-th moment and true 4-th moment, which requires predicting 8-th moment, We continue to use this idea: we estimate 2-th, 4-th, 8-th, ..., $O(\log KH)$-th moments. The index $m$ is used for moments, and $\Lambda_0$ is the index set reserved for moments. We note that the proof in [Zhang et al., 2020a] also depends on the higher moments. The main difference is here we estimate these higher moments explicitly.

# 6 Discussions

By incorporating the variance information in the confidence set construction, we derive the first variance-dependent regret bound for linear bandits and the nearly horizon-free regret bound for linear mixture MDP. Below we discuss limitations of our work and some future directions.

One drawback of our result is that our dependency on $d$ is large. The main reason is our bounds rely on the convex potential lemma (Lemma 17), which is $\tilde{O}(d^4)$. In analogous to the elliptical potential lemma in [Abbasi-Yadkori et al., 2011], we believe that this bound can be improved to $\tilde{O}(d)$. This improvement will directly reduce the dependencies on $d$ in our bounds.

Another drawback is that our method is not computationally efficient. This is a common issue in elimination-based algorithms. We note that the issue of computational tractability is common in sequential decision-making problems [Zhang and Ji, 2019, Wang et al., 2020a, Bartlett and Tewari, 2012, Zanette et al., 2020, Krishnamurthy et al., 2016, Jiang et al., 2017, Sun et al., 2019, Jin et al., 2021, Du et al., 2021, Dong et al., 2020]. We leave it as a future direction to design computationally efficient algorithms that enjoy variance-dependent bounds for settinsg considered in this paper.

Lastly, in this paper, we only study linear function approximation. It would be interesting to generalize the ideas in this paper to other settings with function approximation schemes [Yang and Wang, 2019, Jin et al., 2020, Zanette et al., 2020, Wang et al., 2020c, Russo and Van Roy, 2013, Jiang et al., 2017, Sun et al., 2019, Du et al., 2021, Jin et al., 2021].

# Acknowledgement

Simon S. Du gratefully acknowledges funding from NSF Award's IIS-2110170 and DMS-2134106.

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
