## A Technical Lemmas

**Lemma 6** ([Azuma, 1967]). *Let $(M_n)_{n \geqslant 0}$ be a martingale such that $M_0 = 0$ and $|M_n - M_{n-1}| \leqslant b$ almost surely for every $n \geqslant 1$. Then we have*

$$\Pr\left[|M_n| \geqslant b\sqrt{2n\log(2/\delta)}\right] \leqslant \delta.$$

**Lemma 7** ([Zhang et al., 2020c], Lemma 9). *Let $\{\mathcal{F}_i\}_{i \geqslant 0}$ be a filtration. Let $\{X_i\}_{i \geqslant 1}$ be a real-valued stochastic process adapted to $\{\mathcal{F}_i\}_{i \geqslant 0}$ such that $0 \leqslant X_i \leqslant 1$ almost surely and that $X_i$ is $\mathcal{F}_i$-measurable. For every $\delta \in (0, 1), c \geqslant 1$, we have*

$$\Pr\left[\exists n \geqslant 1 : \sum_{i=1}^{n} \mathbb{E}[X_i \mid \mathcal{F}_{i-1}] \geqslant 4c\ln\frac{4}{\delta}, \sum_{i=1}^{n} X_i \leqslant c\ln\frac{4}{\delta}\right] \leqslant \delta.$$

**Lemma 8.** *Let $\{\mathcal{F}_i\}_{i \geqslant 0}$ be a filtration. Let $\{X_i\}_{i \geqslant 1}$ be a real-valued stochastic process adapted to $\{\mathcal{F}_i\}_{i \geqslant 0}$ such that $0 \leqslant X_i \leqslant 1$ almost surely and that $X_i$ is $\mathcal{F}_i$-measurable. For every $\delta \in (0, 1), c \geqslant 1$, we have*

$$\Pr\left[\exists n \geqslant 1 : \sum_{i=1}^{n} X_i \geqslant 4c\ln\frac{4}{\delta}, \sum_{i=1}^{n} \mathbb{E}[X_i \mid \mathcal{F}_{i-1}] \leqslant c\ln\frac{4}{\delta}\right] \leqslant \delta.$$

*Proof.* We follow the proof of Lemma 9 in [Zhang et al., 2020c]. Let $\lambda > 0$ be a parameter, $\mu_i = \mathbb{E}[X_i \mid \mathcal{F}_{i-1}]$. Define $Y_n = \exp\left(\lambda \sum_{i=1}^{n} X_i - (e^\lambda - 1) \sum_{i=1}^{n} \mu_i\right)$ for $n \geqslant 0$. Note that $\mathbb{E}[e^{\lambda X}] \leqslant \mu e^\lambda + (1 - \mu) \leqslant e^{\mu(e^\lambda - 1)}$, so $\mathbb{E}[e^{\lambda X_i - (e^\lambda - 1)\mu_i} \mid \mathcal{F}_{i-1}] \leqslant 1$, thus $\{Y_n\}_{n \geqslant 0}$ is a super-martingale. Let $\tau = \min\{n : \sum_{i=1}^{n} X_i \geqslant 4c\ln(4/\delta)\}$ be a stopping time, then we have $\left|Y_{\min\{\tau, n\}}\right| \leqslant e^{\lambda(4c\ln(4/\delta)+1)} < +\infty$ almost surely for every $n \geqslant 0$. Therefore, by the optional stopping theorem, we have $\mathbb{E}[Y_\tau] \leqslant 1$. Finally, we have

$$\Pr\left[\exists n \geqslant 1 : \sum_{i=1}^{n} X_i \geqslant 4c\ln\frac{4}{\delta}, \sum_{i=1}^{n} \mu_i \leqslant c\ln\frac{4}{\delta}\right] \leqslant \Pr\left[\sum_{i=1}^{\tau} \mu_i \leqslant c\ln\frac{4}{\delta}\right]$$

$$\leqslant \Pr\left[Y_\tau \geqslant \exp\left(\lambda \sum_{i=1}^{\tau} X_i - (e^\lambda - 1)c\ln\frac{4}{\delta}\right)\right]$$

$$\leqslant \Pr\left[Y_\tau \geqslant \exp\left(\lambda(4c\ln\frac{4}{\delta} - 1) - (e^\lambda - 1)c\ln\frac{4}{\delta}\right)\right]$$

$$\leqslant \exp\left(\lambda(1 - 4c\ln\frac{2}{\delta}) + (e^\lambda - 1)c\ln\frac{4}{\delta}\right)$$

$$= e^\lambda e^{(e^\lambda - 1 - 4\lambda)c\ln(4/\delta)}.$$

Choosing $\lambda = 1$, we have

$$e^\lambda e^{(e^\lambda - 1 - 4\lambda)c\ln(4/\delta)} \leqslant e \cdot e^{-2c\ln(4/\delta)} = e(\frac{\delta}{4})^c \leqslant \frac{e}{4}\delta \leqslant \delta,$$

which concludes the proof. $\qquad\square$

**Lemma 9.** *Let $\{\mathcal{F}_i\}_{i \geqslant 0}$ be a filtration. Let $\{X_i\}_{i=1}^{n}$ be a sequence of random variables such that $|X_i| \leqslant 1$ almost surely, that $X_i$ is $\mathcal{F}_i$-measurable. For every $\delta \in (0, 1)$, we have*

$$\Pr\left[\sum_{i=1}^{n} \mathbb{E}[X_i^2 \mid \mathcal{F}_{i-1}] \geqslant \sum_{i=1}^{n} 8X_i^2 + 4\ln\frac{4}{\delta}\right] \leqslant (\lceil \log_2 n \rceil + 1)\delta.$$

*Proof.* Let $Y = \sum_{i=1}^{n} \mathbb{E}[X_i^2 \mid \mathcal{F}_{i-1}], Z = \sum_{i=1}^{n} X_i^2$. Applying Lemma 7 with the sequence $\{X_i^2\}_{i=1}^{n}$, we have for every $c \geqslant 1$,

$$\Pr\left[Y \geqslant 4c\ln\frac{4}{\delta}, Z \leqslant c\ln\frac{4}{\delta}\right] \leqslant \delta.$$

Therefore, we have

$$\Pr\left[Y \geq 8Z + 4\ln\frac{4}{\delta}\right]$$

$$\leq \sum_{j=1}^{\lceil\log_2 n\rceil} \Pr\left[Y \geq 8Z + 4\ln\frac{4}{\delta}, 2^{j-1}\ln\frac{4}{\delta} \leq Z \leq 2^j\ln\frac{4}{\delta}\right] + \Pr\left[Y \geq 8Z + 4\ln\frac{4}{\delta}, Z \leq \ln\frac{4}{\delta}\right]$$

$$\leq \sum_{j=1}^{\lceil\log_2 n\rceil} \Pr\left[Y \geq 8Z, 2^{j-1}\ln\frac{4}{\delta} \leq Z \leq 2^j\ln\frac{4}{\delta}\right] + \Pr\left[Y \geq 4\ln\frac{4}{\delta}, Z \leq \ln\frac{4}{\delta}\right]$$

$$\leq \sum_{j=1}^{\lceil\log_2 n\rceil} \Pr\left[Y \geq 8\cdot 2^{j-1}\ln\frac{4}{\delta}, 2^{j-1}\ln\frac{4}{\delta} \leq Z \leq 2^j\ln\frac{4}{\delta}\right] + \Pr\left[Y \geq 4\ln\frac{4}{\delta}, Z \leq \ln\frac{4}{\delta}\right]$$

$$\leq \sum_{j=1}^{\lceil\log_2 n\rceil} \Pr\left[Y \geq 4\cdot 2^j\ln\frac{4}{\delta}, Z \leq 2^j\ln\frac{4}{\delta}\right] + \Pr\left[Y \geq 4\ln\frac{4}{\delta}, Z \leq \ln\frac{4}{\delta}\right]$$

$$\leq (\lceil\log_2 n\rceil + 1)\delta$$

as desired. $\qquad\square$

**Lemma 10.** *Let $\{\mathcal{F}_i\}_{i\geq 0}$ be a filtration. Let $\{X_i\}_{i=1}^n$ be a sequence of random variables such that $|X_i| \leq 1$ almost surely, that $X_i$ is $\mathcal{F}_i$-measurable. For every $\delta \in (0,1)$, we have*

$$\Pr\left[\sum_{i=1}^n X_i^2 \geq \sum_{i=1}^n 8\,\mathbb{E}[X_i^2 \mid \mathcal{F}_{i-1}] + 4\ln\frac{4}{\delta}\right] \leq (\lceil\log_2 n\rceil + 1)\delta.$$

*Proof.* Let $Y = \sum_{i=1}^n X_i^2, Z = \sum_{i=1}^n \mathbb{E}[X_i^2 \mid \mathcal{F}_{i-1}]$. Applying Lemma 8 with the sequence $\{X_i^2\}_{i=1}^n$, we have for every $c \geq 1$,

$$\Pr\left[Y \geq 4c\ln\frac{4}{\delta}, Z \leq c\ln\frac{4}{\delta}\right] \leq \delta.$$

Therefore, we have

$$\Pr\left[Y \geq 8Z + 4\ln\frac{4}{\delta}\right]$$

$$\leq \sum_{j=1}^{\lceil\log_2 n\rceil} \Pr\left[Y \geq 8Z + 4\ln\frac{4}{\delta}, 2^{j-1}\ln\frac{4}{\delta} \leq Z \leq 2^j\ln\frac{4}{\delta}\right] + \Pr\left[Y \geq 8Z + 4\ln\frac{4}{\delta}, Z \leq \ln\frac{4}{\delta}\right]$$

$$\leq \sum_{j=1}^{\lceil\log_2 n\rceil} \Pr\left[Y \geq 8Z, 2^{j-1}\ln\frac{4}{\delta} \leq Z \leq 2^j\ln\frac{4}{\delta}\right] + \Pr\left[Y \geq 4\ln\frac{4}{\delta}, Z \leq \ln\frac{4}{\delta}\right]$$

$$\leq \sum_{j=1}^{\lceil\log_2 n\rceil} \Pr\left[Y \geq 8\cdot 2^{j-1}\ln\frac{4}{\delta}, 2^{j-1}\ln\frac{4}{\delta} \leq Z \leq 2^j\ln\frac{4}{\delta}\right] + \Pr\left[Y \geq 4\ln\frac{4}{\delta}, Z \leq \ln\frac{4}{\delta}\right]$$

$$\leq \sum_{j=1}^{\lceil\log_2 n\rceil} \Pr\left[Y \geq 4\cdot 2^j\ln\frac{4}{\delta}, Z \leq 2^j\ln\frac{4}{\delta}\right] + \Pr\left[Y \geq 4\ln\frac{4}{\delta}, Z \leq \ln\frac{4}{\delta}\right]$$

$$\leq (\lceil\log_2 n\rceil + 1)\delta$$

as desired. $\qquad\square$

**Lemma 11** ([Zhang et al., 2020c], Lemma 11)**.** *Let $(M_n)_{n\geq 0}$ be a martingale such that $M_0 = 0$ and $|M_n - M_{n-1}| \leq b$ almost surely for every $n \geq 1$. For each $n \geq 0$, let $\mathcal{F}_n = \sigma(M_0, \ldots, M_n)$ and let $\mathrm{Var}_n = \sum_{i=1}^n \mathbb{E}[(M_i - M_{i-1})^2 \mid \mathcal{F}_{i-1}]$. Then for any $n \geq 1$ and $\epsilon, \delta > 0$, we have*

$$\Pr\left[|M_n| \geq 2\sqrt{2\mathrm{Var}_n \ln(1/\delta)} + 2\sqrt{\epsilon\ln(1/\delta)} + 2b\ln(1/\delta)\right] \leq 2(\log_2(b^2 n/\epsilon) + 1)\delta.$$

**Lemma 12.** *Let $\lambda_1, \lambda_2, \lambda_4 > 0$, $\lambda_3 \geqslant 1$ and $\kappa = \max\{\log_2(\lambda_1), 1\}$. Let $a_1, a_2, \ldots, a_\kappa$ be nonnegative reals such that $a_i \leqslant \lambda_1$ and $a_i \leqslant \lambda_2 \sqrt{a_i + a_{i+1} + 2^{i+1}\lambda_3} + \lambda_4$ for any $1 \leqslant i \leqslant \kappa$ (with $a_{\kappa+1} = \lambda_1$). Then we have that*

$$a_1 \leqslant 22\lambda_2^2 + 6\lambda_4 + 4\lambda_2\sqrt{2\lambda_3}.$$

*Proof.* Note that

$$a_i \leqslant \lambda_2\sqrt{a_i} + \lambda_2\sqrt{a_{i+1} + 2^{i+1}\lambda_3} + \lambda_4,$$

so we have

$$a_i \leqslant \left(\lambda_2 + \sqrt{\lambda_2\sqrt{a_{i+1} + 2^{i+1}\lambda_3} + \lambda_4}\right)^2 \leqslant 2\lambda_2^2 + 2\lambda_2\sqrt{a_{i+1} + 2^{i+1}\lambda_3} + 2\lambda_4.$$

By Lemma 11 in [Zhang et al., 2020a], we have

$$a_1 \leqslant \max\left\{\left(2\lambda_2 + \sqrt{(2\lambda_2)^2 + (2\lambda_2^2 + 2\lambda_4)}\right)^2, 2\lambda_2\sqrt{8\lambda_3} + 2\lambda_2^2 + 2\lambda_4\right\}$$

$$\leqslant \max\{20\lambda_2^2 + 4\lambda_4, 2\lambda_2\sqrt{8\lambda_3} + 2\lambda_2^2 + 2\lambda_4\} \leqslant 22\lambda_2^2 + 6\lambda_4 + 4\lambda_2\sqrt{2\lambda_3},$$

which concludes the proof. $\square$

# B Limitations of Previous Approaches

In the example in Section 1, if we know $x_i \leqslant \sqrt{\frac{1}{K}}$ for $1 \leqslant i \leqslant K$, the best confidence region for $\theta^*$ should be $\Theta_t = \{\theta | \|\theta - \hat{\theta}_t\|_{\Lambda_{t-1}} \leqslant C(\sigma\sqrt{d} + \lambda^{1/2})\}$, and we can obtain a variance-aware regret bound by letting $\lambda = \sigma^2$. However, if we let $x_{K+1} = 1$ and use the same concentration inequality as before, the confidence region would be $\Theta_{K+1} = \{\theta | \|\theta - \hat{\theta}_t\|_{\Lambda_{t-1}}\} \leqslant C(\sigma\sqrt{d} + 1 + \lambda^{1/2})$.

We present the detailed computation as below. Choose $\theta^* = \Theta(1)$. $\theta^* - \hat{\theta}_{K+1} = -\frac{\sum_{i=1}^{K+1} x_i \epsilon_i}{\lambda + \sum_{i=1}^{K+1} x_i^2} + \frac{\lambda\theta^*}{\lambda + \sum_{i=1}^{K+1} x_i^2}$. When $\epsilon_i$ is bounded in $[-1, 1]$ with variance $\sigma^2$, following Bernstein inequality, we have that $\left|\frac{\sum_{i=1}^{K+1} x_i \epsilon_i}{\lambda + \sum_{i=1}^{K+1} x_i^2}\right| \leqslant \frac{\sqrt{\sigma^2 \sum_{i=1}^{K+1} x_i^2} + \max_i x_i}{\lambda + \sum_{i=1}^{K+1} x_i^2}$. Therefore, the best confidence interval we have is

$$\|\theta^* - \hat{\theta}_{K+1}\|_{\Lambda_K} \lesssim \sqrt{\frac{\sigma^2 \sum_{i=1}^{K+1} x_i^2}{\lambda + \sum_{i=1}^{K+1} x_i^2}} + \frac{\max_i x_i}{\sqrt{\lambda + \sum_{i=1}^{K+1} x_i^2}} + \frac{\lambda\theta^*}{\sqrt{\lambda + \sum_{i=1}^{K+1} x_i^2}}$$

$$= \Theta\left(\sqrt{\frac{\sigma^2}{\lambda + 1}} + \frac{1 + \lambda}{\sqrt{1 + \lambda}}\right),$$

i.e., $|\theta^* - \hat{\theta}_{K+1}|_{\Lambda_K} \lesssim \Theta(\sigma + \lambda^{1/2} + 1)$. Therefore, to maintain a confidence region for the general case following methods in [Zhou et al., 2020a, Faury et al., 2020], the term $1 + \lambda^{1/2}$ is unavoidable. This counter example highlights the necessity of our peeling step in the algorithm.

**Remark 4.** *We note that for $\sigma$-sub-Gaussian noise (instead of $\sigma^2$ variance and 1-sub-Gaussian), one can ensure that $\left|\frac{\sum_{i=1}^{K+1} x_i \epsilon_i}{\lambda + \sum_{i=1}^{K+1} x_i^2}\right| \leqslant \frac{\sqrt{\sigma^2 \sum_{i=1}^{K+1} x_i^2}}{\lambda + \sum_{i=1}^{K+1} x_i^2}$, which help to reduce the width of confidence interval and obtain $|\theta^* - \hat{\theta}_{K+1}|_{\Lambda_K} \leqslant O(\sigma + \lambda^{1/2})$.*

# C Proof of Lemma 1

In this section, we present the proof of Lemma 1.

**Restatement of Lemma 1** *Let $f(x) \geqslant 0$ be a convex function over $\mathbb{R}$ such that $\frac{f(x)}{x^2} \leqslant \frac{f(y)}{y^2} \leqslant 1$ and $f(x) \geqslant f(y)$ if $x^2 \geqslant y^2 > 0$. Fix $\ell \in (0, 1]$. For any $\boldsymbol{x}_1, \boldsymbol{x}_2, \ldots, \boldsymbol{x}_t \in \mathbb{B}_2^d(1)$ and $\boldsymbol{\mu}_1, \boldsymbol{\mu}_2, \ldots, \boldsymbol{\mu}_t \in \mathbb{B}_2^d(1)$, we have that*

$$\sum_{i=1}^{t} \min\left\{ \frac{f(\boldsymbol{x}_i \boldsymbol{\mu}_i)}{\sum_{j=1}^{i-1} f(\boldsymbol{x}_j \boldsymbol{\mu}_i) + \ell^2}, 1 \right\} \leqslant O(d^4 \log(Cdt/\ell)). \tag{8}$$

Let $f(x)$ and $\ell$ be fixed. To prove Lemma 1, we have the lemmas below.

**Lemma 13.** *For any $\boldsymbol{x}_1.\boldsymbol{x}_2, \ldots, \boldsymbol{x}_t \in \mathbb{B}_2^d(1)$ and $\boldsymbol{\mu}_1, \boldsymbol{\mu}_2, \ldots, \boldsymbol{\mu}_n \in \mathbb{B}_2^d(1)$, we have that*

$$\sum_{i=1}^{t} \min\left\{ \frac{f(\boldsymbol{x}_i \boldsymbol{\mu}_i)}{\sum_{j=1}^{t} f(\boldsymbol{x}_j \boldsymbol{\mu}_i) + \ell^2}, 1 \right\} \leqslant O(d \log(Cdt/\ell)). \tag{9}$$

**Lemma 14.** *Let $\boldsymbol{x}_1, \boldsymbol{x}_2, \ldots, \boldsymbol{x}_t \in \mathbb{B}_2^d(1)$ be a sequence of vectors. If there exists a sequence $0 = \tau_0 < \tau_1 < \tau_2 < \ldots < \tau_z = t$ such that for each $1 \leqslant \zeta \leqslant z$, there exists $\boldsymbol{\mu}_\zeta \in \mathbb{B}_2^d(1)$ such that*

$$\sum_{i=1}^{\tau_\zeta} f(\boldsymbol{x}_i \boldsymbol{\mu}_\zeta) + \ell^2 > 4(d+2)^2 \times \left( \sum_{i=1}^{\tau_{\zeta-1}} f(\boldsymbol{x}_i \boldsymbol{\mu}_\zeta) + \ell^2 \right), \tag{10}$$

*then $z \leqslant O(d \log^2(dt/\ell))$.*

We present the proofs of Lemma 13 and 14 respectively in Section C.1 and C.2. Given these two lemmas, we continue analysis as below.

Let $\tau_0 = 0$ and for $i \geqslant 1$, we let

$$\tau_i = \min\{t + 1\} \cup \left\{ \tau \,\middle|\, \exists \tau_{i-1} \leqslant \tau' < \tau, \sum_{j=1}^{\tau} f(\boldsymbol{x}_j \boldsymbol{\mu}_{\tau'}) + \ell^2 > 4(d+2)^2 \left( \sum_{j=1}^{\tau'} f(\boldsymbol{x}_j \boldsymbol{\mu}_{\tau'}) + \ell^2 \right) \right\}.$$

Let $k = \min\{i \mid \tau_i = t + 1\}$. Then $k$ is well-defined and $k \leqslant O(d \log^2(dt))$ by Lemma 14. Furthermore, for any $\kappa < k$ and any $\tau_\kappa \leqslant i_1 < i_2 < \tau_{\kappa+1}$, we have

$$\sum_{j=1}^{i_2} f(\boldsymbol{x}_j \boldsymbol{\mu}_{i_1}) + \ell^2 \leqslant 4(d+2)^2 \left( \sum_{j=1}^{i_1} f(\boldsymbol{x}_j \boldsymbol{\mu}_{i_1}) + \ell^2 \right). \tag{11}$$

Now we are ready to prove Lemma 1. We have

$$\sum_{i=1}^{t} \min\left\{ \frac{f(\boldsymbol{x}_i \boldsymbol{\mu}_i)}{\sum_{j=1}^{i-1} f(\boldsymbol{x}_j \boldsymbol{\mu}_i) + \ell^2}, 1 \right\} \leqslant 2 \sum_{i=1}^{t} \frac{f(\boldsymbol{x}_i \boldsymbol{\mu}_i)}{\sum_{j=1}^{i} f(\boldsymbol{x}_j \boldsymbol{\mu}_i) + \ell^2}$$

$$\leqslant 8(d+2)^2 \sum_{\kappa=1}^{k} \left( \sum_{i=\tau_{\kappa-1}}^{\tau_\kappa - 1} \frac{f(\boldsymbol{x}_i \boldsymbol{\mu}_i)}{\sum_{j=1}^{\tau_\kappa - 1} f(\boldsymbol{x}_j \boldsymbol{\mu}_i) + \ell^2} \right) \tag{12}$$

$$\leqslant 8(d+2)^2 \sum_{\kappa=1}^{k} \left( \sum_{i=\tau_{\kappa-1}}^{\tau_\kappa - 1} \frac{f(\boldsymbol{x}_i \boldsymbol{\mu}_i)}{\sum_{j=\tau_{\kappa-1}}^{\tau_\kappa - 1} f(\boldsymbol{x}_j \boldsymbol{\mu}_i) + \ell^2} \right),$$

$$\leqslant k \times O(d^2) \times O(d \log(t/\ell)) \leqslant O(d^4 \log^3(dt)), \tag{13}$$

where (12) uses (11) and (13) uses Lemma 13.

## C.1   Proof of Lemma 13

**Restatement of Lemma 13** *For any $\boldsymbol{x}_1.\boldsymbol{x}_2, \ldots, \boldsymbol{x}_t \in \mathbb{B}_2^d(1)$ and $\boldsymbol{\mu}_1, \boldsymbol{\mu}_2, \ldots, \boldsymbol{\mu}_n \in \mathbb{B}_2^d(1)$, we have that*

$$\sum_{i=1}^{t} \min\left\{ \frac{f(\boldsymbol{x}_i \boldsymbol{\mu}_i)}{\sum_{j=1}^{t} f(\boldsymbol{x}_j \boldsymbol{\mu}_i) + \ell^2}, 1 \right\} \leqslant O(d \log(Cdt/\ell)). \tag{14}$$

*Proof.* Let $S_t$ be the permutation group over $[t]$. We claim that if

$$\sum_{i=1}^{t} \frac{f\boldsymbol{x}_i\boldsymbol{\mu}_i)}{\sum_{j=1}^{t} f(\boldsymbol{x}_j\boldsymbol{\mu}_i) + \ell^2} = \max_{\xi \in S_t} \sum_{i=1}^{t} \frac{f(\boldsymbol{x}_{\xi(i)}\boldsymbol{\mu}_i)}{\sum_{j=1}^{t} f2(\boldsymbol{x}_{\xi(j)}\boldsymbol{\mu}_i,) + \ell^2}, \tag{15}$$

then there exists some $i$ such that $(\boldsymbol{x}_i\boldsymbol{\mu}_i)^2 \geqslant (\boldsymbol{x}_j\boldsymbol{\mu}_i)^2$ for any $j \in [t]$. Otherwise, we construct a directed graph $G = (V, E)$ where $V = [t]$ and edge $(i, j)$ with $i \neq j$ is in $E$ if and only if $(\boldsymbol{x}_j\boldsymbol{\mu}_i)^2 \geqslant (x_{j'}\boldsymbol{\mu}_i)^2$ for any $j' \in [t]$. Let $d(i)$ be the out degree of $i$. By assuming $\{(\boldsymbol{x}_i\boldsymbol{\mu}_i)^2 \geqslant (\boldsymbol{x}_j^\top\boldsymbol{\mu}_i)^2, \forall j \in [t]\}$ fails to hold, we learn that $d(i) \geqslant 1$ for every $i$, so there exists a circle $(i_1, i_2, \ldots, i_k)$ in $G$. Consider the permutation $\xi$ such that $\xi(i_j) = i_{j+1}$ for $j \in [k]$ (with $i_{k+1} := i_1$) and $\xi(i) = i$ for $i \notin \{i_1, \ldots, i_k\}$. By definition, we have $(\boldsymbol{\mu}_{i_j}\boldsymbol{x}_{\xi(i_j)})^2 > (\boldsymbol{\mu}_{i_j}\boldsymbol{x}_{i_j})^2$ for $j \in [k]$, which implies that $f(\boldsymbol{\mu}_{i_j}\boldsymbol{x}_{\xi(i_j)}) > f(\boldsymbol{\mu}_{i_j}\boldsymbol{x}_{i_j})$ for $j \in [k]$. Therefore

$$\sum_{i=1}^{t} \frac{f(\boldsymbol{x}_i\boldsymbol{\mu}_i)}{\sum_{j=1}^{t} f(\boldsymbol{x}_j\boldsymbol{\mu}_i) + \ell^2} < \sum_{i=1}^{t} \frac{f(\boldsymbol{x}_{\xi(i)}\boldsymbol{\mu}_i)}{\sum_{j=1}^{t} f(\boldsymbol{x}_j\boldsymbol{\mu}_i) + \ell^2} = \sum_{i=1}^{t} \frac{f(\boldsymbol{x}_{\xi(i)}\boldsymbol{\mu}_i)}{\sum_{j=1}^{t} f(\boldsymbol{x}_{\xi(j)}\boldsymbol{\mu}_i) + \ell^2},$$

which leads to contradiction.

We assume that (15) holds, otherwise we can bound an upper bound of the original quantity. Therefore, we can find an index $i$ such that $(\boldsymbol{x}_i\boldsymbol{\mu}_i)^2 \geqslant (\boldsymbol{x}_j^\top\boldsymbol{\mu}_i)^2$ for any $j \in [t]$. Without loss of generality, we assume $i = 1$. Because $\frac{f(x)}{x^2}$ is decreasing in $x$, so we have

$$\frac{f(\boldsymbol{x}_1\boldsymbol{\mu}_1)}{(\boldsymbol{x}_1\boldsymbol{\mu}_1)^2} \leqslant \frac{f(\boldsymbol{x}_j\boldsymbol{\mu}_1)}{(\boldsymbol{x}_j\boldsymbol{\mu}_1)^2}$$

for any $j \in [t]$, which implies

$$\frac{f(\boldsymbol{x}_1\boldsymbol{\mu}_1)}{\sum_{j=1}^{t} f(\boldsymbol{x}_j\boldsymbol{\mu}_1) + \ell^2} = \frac{(\boldsymbol{x}_1\boldsymbol{\mu}_1)^2}{\left(\sum_{j=1}^{t} f(\boldsymbol{x}_j\boldsymbol{\mu}_1) + \ell^2\right) \cdot \frac{(\boldsymbol{x}_1\boldsymbol{\mu}_1)^2}{f(\boldsymbol{x}_1\boldsymbol{\mu}_1)}} \leqslant \frac{(\boldsymbol{x}_1\boldsymbol{\mu}_1)^2}{\sum_{j=1}^{t} (\boldsymbol{x}_j\boldsymbol{\mu}_1)^2 + \ell^2}. \tag{16}$$

Therefore, we have

$$\sum_{i=1}^{t} \frac{f(\boldsymbol{x}_i\boldsymbol{\mu}_i)}{\sum_{j=1}^{t} f(\boldsymbol{x}_i\boldsymbol{\mu}_i) + \ell^2} \leqslant \frac{(\boldsymbol{x}_1\boldsymbol{\mu}_1)^2}{\sum_{j=1}^{t} (\boldsymbol{x}_j\boldsymbol{\mu}_1)^2 + \ell^2} + \sum_{i=2}^{t} \frac{f(\boldsymbol{x}_i\boldsymbol{\mu}_i)}{\sum_{j=1}^{t} f(\boldsymbol{x}_i\boldsymbol{\mu}_i) + \ell^2}$$

$$\leqslant \frac{(\boldsymbol{x}_1\boldsymbol{\mu}_1)^2}{\sum_{j=1}^{t} (\boldsymbol{x}_j\boldsymbol{\mu}_1)^2 + \ell^2} + \sum_{i=2}^{t} \frac{f(\boldsymbol{x}_i\boldsymbol{\mu}_i)}{\sum_{j=2}^{t} f(\boldsymbol{x}_i\boldsymbol{\mu}_i) + \ell^2}. \tag{17}$$

Similarly, we can show that there exists a permutation $\xi^* \in S_t$ such that

$$\sum_{i=1}^{t} \frac{f(\boldsymbol{x}_i\boldsymbol{\mu}_i)}{\sum_{j=1}^{t} f(\boldsymbol{x}_j\boldsymbol{\mu}_i) + \ell^2} \leqslant \sum_{i=1}^{t} \frac{(\boldsymbol{x}_{\xi^*(i)}^\top\boldsymbol{\mu}_i)^2}{\sum_{j=i}^{t} (\boldsymbol{x}_{\xi^*(j)}^\top\boldsymbol{\mu}_i)^2 + \ell^2}. \tag{18}$$

Finally, by Lemma 15, we have that

$$\sum_{i=1}^{t} \frac{(\boldsymbol{x}_{\xi^*(i)}\boldsymbol{\mu}_i)^2}{\sum_{j=i}^{t} (\boldsymbol{x}_{\xi^*(j)}\boldsymbol{\mu}_i)^2 + \ell^2} = \sum_{i=1}^{t} \min\left\{ \frac{(\boldsymbol{x}_{\xi^*(i)}\boldsymbol{\mu}_i)^2}{\sum_{j=i}^{t} (x_{\xi^*(j)}\boldsymbol{\mu}_i)^2 + \ell^2}, 1 \right\} \leqslant O(d\log(t/\ell)).$$

$\square$

## C.2 Proof of Lemma 14

**Restatement of Lemma 14** *Let $\boldsymbol{x}_1, \boldsymbol{x}_2, \ldots, \boldsymbol{x}_t \in \mathbb{B}_2^d(1)$ be a sequence of vectors. If there exists a sequence $0 = \tau_0 < \tau_1 < \tau_2 < \ldots < \tau_z = t$ such that for each $1 \leqslant \zeta \leqslant z$, there exists $\boldsymbol{\mu}_\zeta \in \mathbb{B}_2^d(1)$ such that*

$$\sum_{i=1}^{\tau_\zeta} f(\boldsymbol{x}_i\boldsymbol{\mu}_\zeta) + \ell^2 > 4(d+2)^2 \times \left( \sum_{i=1}^{\tau_{\zeta-1}} f(\boldsymbol{x}_i\boldsymbol{\mu}_\zeta) + \ell^2 \right), \tag{19}$$

*then $z \leqslant O(d\log^2(dt/\ell))$.*

*Proof.* If $f(1) \leqslant \ell^2/t$, then the conclusion holds trivially because $0 \leqslant f(x) \leqslant f(1) \leqslant \ell^2/t$ for all $x \in [-1, 1]$. Suppose $f(1) > \ell^2/t$. Since $\frac{f(x)}{x^2} \leqslant \frac{f(y)}{y^2} \leqslant 1$ for all $x^2 \geqslant y^2$, we have that for $0 < \lambda \leqslant 1$ and any $x \in \mathbb{R}$, $f(\lambda x) \geqslant \lambda^2 f(x)$.

Let $\boldsymbol{e}_i = [0, \ldots, 1, \ldots, 0]$ be the one-hot vector whose only 1 entry is at its $i$-th coordinate. Noting that $f(x) \leqslant x^2$, $|\boldsymbol{x}_i \boldsymbol{\mu}_\zeta| \leqslant \|\boldsymbol{\mu}_\zeta\|_2$ and

$$\sum_{i=1}^{\tau_\zeta} f(\boldsymbol{x}_i \boldsymbol{\mu}_\zeta) > 4(d+2)^2 \times \left( \sum_{i=1}^{\tau_{\zeta-1}} f(\boldsymbol{x}_i \boldsymbol{\mu}_\zeta) + \ell^2 \right) - \ell^2 \geqslant 4d^2 \ell^2$$

we have that $|\boldsymbol{\mu}_\zeta|_2 \geqslant \sqrt{\frac{4d^2\ell^2}{t}}$. Define $E_\tau(\boldsymbol{\mu}) = \sum_{i=1}^t f(\boldsymbol{x}_t \boldsymbol{\mu}) + \frac{\ell^2}{d} \sum_{i=1}^d f(\boldsymbol{e}_i \boldsymbol{\mu})$. Then $E_\tau(\boldsymbol{\mu})$ is convex in $\boldsymbol{\mu}$ because $f(x)$ is convex in $x$. By definition, we have that

$$E_\tau(\boldsymbol{\mu}) \leqslant \sum_{i=1}^\tau f(\boldsymbol{x}_i \boldsymbol{\mu}) + \ell^2.$$

By (19), we have that

$$E_{\tau_\zeta}(\boldsymbol{\mu}_\zeta) \geqslant \sum_{i=1}^{\tau_\zeta} f(\boldsymbol{x}_i \boldsymbol{\mu}_\zeta) \geqslant 4d^2 \left( \sum_{i=1}^{\tau_{\zeta-1}} f(\boldsymbol{x}_i \boldsymbol{\mu}_\zeta) + \ell^2 \right) \geqslant 4d^2 E_{\tau_{\zeta-1}}(\boldsymbol{\mu}_\zeta). \tag{20}$$

Define

$$\Lambda = \left\{ i \in \mathbb{Z} : \left\lfloor \log_2(d\ell^4/t^2) + 2 \right\rfloor \leqslant i \leqslant 2 \left\lfloor \log_2 t + 2 \right\rfloor \right\}.$$

We consider the convex set $D_{\tau,i} = \{\boldsymbol{\mu} : E_\tau(\mu) \leqslant 2^i\}$ for $i \in \Lambda$. Let $\zeta$ be fixed. Because $\|\boldsymbol{\mu}_\zeta\| \geqslant \sqrt{\frac{4d^2\ell^2}{t}}$ and $\sup_i f(\boldsymbol{e}_i \boldsymbol{\mu}) \geqslant \frac{4d\ell^2}{t} \cdot f(1) \geqslant \frac{4d\ell^4}{t^2}$, we have that $\frac{4d\ell^4}{t^2} \leqslant E_\tau(\boldsymbol{\mu}_\zeta) \leqslant t + \ell^2 \leqslant t + 1$ for any $1 \leqslant \tau \leqslant t$. Then we can find $i_\zeta \in \Lambda$ such that $E_{\tau_{\zeta-1}}(\boldsymbol{\mu}_\zeta) \in (2^{i_\zeta - 1}, 2^{i_\zeta}]$, which means that $\boldsymbol{\mu}_\zeta \in D_{\tau_{\zeta-1}, i_\zeta}$. Note that for $0 \leqslant \lambda \leqslant 1$, $f(\lambda x) \geqslant \lambda^2 f(x)$ for any $x$, it then follows that $E_t(\lambda \boldsymbol{\mu}) \geqslant \lambda^2 E_t(\boldsymbol{\mu})$ for any $t, \boldsymbol{\mu}$. Choosing $\lambda = \frac{1}{d}$, we have that $E_{\tau_\zeta}(\frac{\boldsymbol{\mu}_\zeta}{d}) \geqslant \frac{1}{d^2} E_{\tau_\zeta}(\boldsymbol{\mu}_\zeta) \geqslant 4 E_{\tau_{\zeta-1}}(\boldsymbol{\mu}_\zeta) \geqslant 2^{i_\zeta}$. Therefore, $\frac{\boldsymbol{\mu}_\zeta}{d} \notin D_{\tau_\zeta, i_\zeta}$. In words, the intercept of $D_{\tau_\zeta, i_\zeta}$ in the direction $\boldsymbol{\mu}_\zeta$ is at most $1/d$ times of that of $D_{\tau_{\zeta-1}, i_\zeta}$.

Note that $D_{t,i}$ is decreasing in $t$ for any $i$, so by Lemma 16, we have

$$\text{Volume}(D_{\tau_\zeta, i_\zeta}) \leqslant \frac{6}{7} \text{Volume}(D_{\tau_{\zeta-1}, i_\zeta}).$$

Also note that $\text{Volume}(D_{0,i}) \leqslant (\frac{2t}{\ell})^d$ and $\text{Volume}(D_{t,i}) \geqslant (\frac{1}{dt^3})^d$, so we conclude that $z \leqslant d|\Lambda| \log_{7/6}(2dt^4/\ell) \leqslant O(d \log^2(td/\ell))$.

$\square$

## C.3 Other Lemmas and Proofs

**Lemma 15.** *Fix $\ell \in (0, 1]$. Let $\boldsymbol{x}_1, \boldsymbol{x}_2, \ldots, \boldsymbol{x}_t \in \mathbb{B}_2^d(1)$ and $\boldsymbol{\mu}_1, \boldsymbol{\mu}_2, \ldots, \boldsymbol{\mu}_t \in \mathbb{B}_2^d(1)$ be two sequences of vectors. Then we have*

$$\sum_{i=1}^t \mathbb{I}\left\{ (\boldsymbol{x}_i \boldsymbol{\mu}_i)^2 > \sum_{j=1}^{i-1} (\boldsymbol{x}_j \boldsymbol{\mu}_i)^2 + \ell^2 \right\} \leqslant \sum_{i=1}^t \min\left\{ \frac{(\boldsymbol{x}_i \boldsymbol{\mu}_i)^2}{\sum_{j=1}^{i-1}(\boldsymbol{x}_j \boldsymbol{\mu}_i)^2 + \ell^2}, 1 \right\} \leqslant O(d \log \frac{t}{\ell}). \tag{21}$$

*Proof.* The first inequality in (21) holds clearly. To prove the second inequality, we define $\boldsymbol{U}_0 = \ell^2 \boldsymbol{I}$ and $\boldsymbol{U}_i = \ell^2 \boldsymbol{I} + \sum_{j=1}^i \boldsymbol{x}_j \boldsymbol{x}_j^\top$ for $i \geqslant 1$. Note that

$$\frac{(\boldsymbol{x}_i \boldsymbol{\mu}_i)^2}{\sum_{j=1}^{i-1}(\boldsymbol{x}_j \boldsymbol{\mu}_i)^2 + \ell^2} \leqslant \frac{(\boldsymbol{x}_i \boldsymbol{\mu}_i)^2}{\boldsymbol{\mu}_i^\top \boldsymbol{U}_{i-1} \boldsymbol{\mu}_i} \leqslant \boldsymbol{x}_i^\top \boldsymbol{U}_{i-1}^{-1} \boldsymbol{x}_i,$$

where the first inequality is because $\|\boldsymbol{\mu}_i\|_2 \leqslant 1$ and the second inequality uses the Cauchy's inequality, so we have

$$\sum_{i=1}^{t} \min \left\{ \frac{(\boldsymbol{x}_i \boldsymbol{\mu}_i)^2}{\sum_{j=1}^{i-1}(\boldsymbol{x}_j \boldsymbol{\mu}_i)^2 + \ell^2}, 1 \right\} \leqslant \sum_{i=1}^{t} \min \left\{ \boldsymbol{x}_i^\top \boldsymbol{U}_{i-1}^{-1} \boldsymbol{x}_i, 1 \right\} \leqslant 2d \ln\left(t/\ell^2\right) \leqslant 4d \ln(t/\ell),$$

where the second-to-third inequality uses the elliptical potential lemma. $\square$

**Lemma 16.** *Given $x \in \mathbb{R}^d$, we use $(u(x), l(x))$ to denote the polar coordinate of $x$ where $\|\mu(x)\|_2 = \frac{x}{\|x\|_2}$ is the direction and $l(x) = \|x\|_2$. We also use $(u, \ell)$ to denote the unique element $x$ in $\mathbb{R}^d$ such that $(u(x), l(x)) = (u, \ell)$. Let $D$ be a bounded symmetric convex subset of $\mathbb{R}^d$ with $d \geqslant 2$. Given any direction $\mu \in \partial \mathbb{B}_d$, there exists a unique $l(u) \in \mathbb{R}$ such that $(u, l(u)), (-u, l(u)) \in \partial D$ are on its boundary. Let $D'$ be a bounded symmetric convex subset of $\mathbb{R}^d$ containing $D \subseteq D'$ such that $(u, d \cdot l(u)) \in D'$ for some direction $u \in \partial \mathbb{B}_d$. Then we have that*

$$\mathrm{Volume}(D') \geqslant \frac{7}{6} \mathrm{Volume}(D).$$

*Proof.* Let $A = (u, l(u))$ and $B = (u, d \cdot l(u))$. Since $A$ is on the boundary of $D$, we can find a hyperplane $h_1$ such that $A \in h_1$ and $h_1$ is tangent to $D$. Let $h_2$ be the parallel hyperplane of $h_1$ containing the origin $O \in h_2$. Define

$$H = \left\{ x \in \mathbb{R}^d \,\middle|\, d(x, h_1) + d(x, h_2) = d(h_1, h_2), \exists y \in D, \lambda \in \mathbb{R}, (B - y) = \lambda(B - x) \right\}$$

It is obvious that $\mathrm{Volume}(H) \geqslant \frac{1}{2}\mathrm{Volume}(D)$ since for each $x \in D$ lying between $h_1$ and $h_2$, $x \in H$. Define

$$U = \left\{ x \in \mathbb{R}^d \,\middle|\, d(x, h_2) = d(x, h_1) + d(h_1, h_2), \exists y \in H, \lambda \in [0, 1], x = \lambda y + (1 - \lambda)B \right\}.$$

We claim that

$$\mathrm{Volume}(U) = \left(1 - \frac{1}{d}\right)^d \mathrm{Volume}(U \cup H) = \left(1 - \frac{1}{d}\right)^d \left(\mathrm{Volume}(U) + \mathrm{Volume}(H)\right). \quad (22)$$

To see the first equality, we note that $U$ and $U \cup H$ are both $d$-dimensional pyramids. It then follows from the volume formula and the relation $d(B, O) = d \times d(A, O)$. The second equality is because by their definitions, $U, H$ are separated by the hyperplane $h_1$, and thus they are disjoint. Finally, by (22), we have

$$\mathrm{Volume}(D') \geqslant \mathrm{Volume}(U) + \mathrm{Volume}(H) = (1 + \frac{1}{1 - (1 - 1/d)^d})\mathrm{Volume}(H)$$

$$\geqslant \frac{1}{2}(1 + \frac{1}{(1 - (1 - 1/d)^d)})\mathrm{Volume}(D) \geqslant \frac{7}{6}\mathrm{Volume}(D).$$

$\square$

# D  Missing Proofs in Section 4

## D.1  Application of the General Potential Lemma

As an application of Lemma 1 on linear bandit and linear RL, we have the lemma as below

**Lemma 17.** *Fix $\ell \in (0, 1]$. Let $\boldsymbol{x}_1, \boldsymbol{x}_2, \ldots, \boldsymbol{x}_t \in \mathbb{B}_2^d(1)$ be a sequence of vectors, and $\boldsymbol{\mu}_1, \boldsymbol{\mu}_2, \ldots, \boldsymbol{\mu}_t \in \mathbb{B}_2^d(1)$ be another sequence of vectors. Then we have*

$$\sum_{i=1}^{t} \frac{\mathsf{clip}^2(\boldsymbol{x}_i \boldsymbol{\mu}_i, \ell)}{\sum_{j=1}^{i-1} \mathsf{clip}(\boldsymbol{x}_j \boldsymbol{\mu}_i, \ell)\boldsymbol{x}_j^\top \boldsymbol{\mu}_i + \ell^2} \leqslant O(d^4 \log^3(dt)). \quad (23)$$

*Proof.* Let

$$f_\ell(x) = \begin{cases} x^2, & |x| \leqslant \ell, \\ 2\ell x - \ell^2, & x > \ell, \\ -2\ell x - \ell^2, & x < -\ell \end{cases}$$

be a convex relaxation of the function $x \mapsto \mathsf{clip}(x, \ell)x$. It is easy to see that $f_\ell(x)$ is convex in $x$ and for any $x \in \mathbb{R}, \ell > 0$,

$$\mathsf{clip}(x, \ell)x \leqslant f_\ell(x) \leqslant 2\mathsf{clip}(x, \ell)x \leqslant 2x^2. \tag{24}$$

Let $h(x) = \frac{f_\ell(x)}{2}$. It is easy to see that if $x^2 \geqslant y^2$, $\frac{h(x)}{x^2} = \frac{\mathsf{clip}(x,l)}{2x} \leqslant \frac{\mathsf{clip}(y,l)}{2y} = \frac{h(y)}{y^2} \leqslant 1$. By Lemma 1 with $f(x) = h(x)$, we have that

$$\sum_{i=1}^{t} \frac{h(\boldsymbol{x}_i\boldsymbol{\mu}_i)}{\sum_{j=1}^{i-1} h(\boldsymbol{x}_j\boldsymbol{\mu}_i) + \ell^2} \leqslant O(d^4 \log^3(dt)).$$

By (24), we obtain that

$$\sum_{i=1}^{t} \frac{\mathsf{clip}^2(\boldsymbol{x}_i\boldsymbol{\mu}_i, \ell)}{\sum_{j=1}^{i-1} \mathsf{clip}(\boldsymbol{x}_j\boldsymbol{\mu}_i, \ell)\boldsymbol{x}_j^\top\boldsymbol{\mu}_i + \ell^2} \leqslant \sum_{i=1}^{t} \frac{\mathsf{clip}(\boldsymbol{x}_i\boldsymbol{\mu}_i, \ell)\boldsymbol{x}_i\boldsymbol{\mu}_I}{\sum_{j=1}^{i-1} \mathsf{clip}(\boldsymbol{x}_j\boldsymbol{\mu}_i, \ell)\boldsymbol{x}_j^\top\boldsymbol{\mu}_i + \ell^2}$$

$$\leqslant 4 \sum_{i=1}^{t} \frac{h(\boldsymbol{x}_i\boldsymbol{\mu}_i)}{\sum_{j=1}^{i-1} h(\boldsymbol{x}_j\boldsymbol{\mu}_i) + \ell^2}$$

$$\leqslant O(d^4 \log^3(dt)).$$

The proof is completed. $\qquad\square$

## D.2 Proof of Theorem 3

### D.2.1 Optimism

The equation (1) accounts for the main novelty of our algorithm. We note that our confidence set is different from all previous ones [Dani et al., 2008, Abbasi-Yadkori et al., 2011]. Our confidence set is built based on the following new inequality, which may be of independent interest.

With Lemma 4 in hand, we can easily prove that the optimal $\boldsymbol{\theta}^*$ is always in our confidence set with high probability. The proof details can be found in Appendix D.3.

**Lemma 18.** *With probability at least* $1 - O(\delta \log K)$, *we have* $\boldsymbol{\theta}^* \in \Theta_k$ *for all* $k \in [K]$.

### D.2.2 Bounding the Regret

We bound the regret under the event specified in Lemma 18. We have

$$\mathfrak{R}^K = \sum_{k=1}^{K} (\max_{\boldsymbol{x} \in \mathcal{A}_k} \boldsymbol{x}\boldsymbol{\theta}^* - \boldsymbol{x}_k\boldsymbol{\theta}^*)$$

$$\leqslant \sum_{k=1}^{K} \left( \max_{\boldsymbol{x} \in \mathcal{A}_k, \boldsymbol{\theta} \in \Theta_k} \boldsymbol{x}\boldsymbol{\theta} - \boldsymbol{x}_k\boldsymbol{\theta}^* \right) \leqslant \sum_{k=1}^{K} \boldsymbol{x}_k (\boldsymbol{\theta}_k - \boldsymbol{\theta}^*) = \sum_{k} \boldsymbol{x}_k\boldsymbol{\mu}_k,$$

where second inequality follows from Lemma 18. Therefore, it suffices to bound $\sum_k \boldsymbol{x}_k\boldsymbol{\mu}_k$, for which we have the following lemma.

**Lemma 19.** *With probability* $1 - O(\delta \log K)$, *we have*

$$\sum_{k} \boldsymbol{x}_k\boldsymbol{\mu}_k \leqslant O\left(d^{4.5}(\log^4 dK)(\log \frac{dK}{\delta})\left(\sqrt{d} + \sqrt{\sum_{k=1}^{K} \sigma_k^2}\right)\right).$$

Since this lemma is one of our main technical contribution, we provide more proof details.

*Proof.* First, we define the desired event $\mathcal{E} = \mathcal{E}_1 \cap \mathcal{E}_2$, where

$$\mathcal{E}_1 = \{\forall k \in [K] : \boldsymbol{\theta}^* \in \Theta_k\}, \qquad \mathcal{E}_2 = \left\{ \sum_{k=1}^K \eta_k(\boldsymbol{\theta}^*) \leqslant \sum_{k=1}^K 8\sigma_k^2 + 4\ln\frac{4}{\delta} \right\}.$$

By Lemma 18, we have $\Pr[\mathcal{E}_1] \geqslant 1 - O(\delta)$. By Lemma 10, we have $\Pr[\mathcal{E}_2] \geqslant 1 - O(\delta \log K)$. Therefore, by union bound, we have $\Pr[\mathcal{E}] \geqslant 1 - O(\delta \log K)$.

Now we bound $\sum_k \boldsymbol{x}_k \boldsymbol{\mu}_k$ under the event $\mathcal{E}$ to prove the lemma. Recall that

$$\Phi_k^j(\boldsymbol{\mu}) = \sum_{v=1}^{k-1} \mathsf{clip}_j(\boldsymbol{x}_v\boldsymbol{\mu})\boldsymbol{x}_v\boldsymbol{\mu} + \ell_j^2, \qquad \Psi_k^j(\boldsymbol{\mu}) = \sum_{v=1}^{k-1} \mathsf{clip}_j^2(\boldsymbol{x}_v\boldsymbol{\mu})\eta_v(\boldsymbol{\theta}^*). \tag{25}$$

Recall the definition of $\{\mathcal{K}_j\}_{j=1}^{L_2+1}$ in Section 4.2

To proceed, we need the following claim.

**Claim 20.** *We have*

$$\sum_k \boldsymbol{x}_k \boldsymbol{\mu}_k = \sum_{k \in \mathcal{K}_{L_2+1}} \boldsymbol{x}_k \boldsymbol{\mu}_k + \sum_{j=1}^{L_2} \sum_{k \in \mathcal{K}_j} \boldsymbol{x}_k \boldsymbol{\mu}_k$$

$$\leqslant 1 + \sum_{j=1}^{L_2} \sum_{k \in \mathcal{K}_j} \boldsymbol{x}_k \boldsymbol{\mu}_k \times \frac{3\sqrt{\Psi_k^j(\boldsymbol{\mu}_k)\iota} + \sqrt{\sum_{v=1}^{k-1} 2\mathsf{clip}_j^2(\boldsymbol{x}_v\boldsymbol{\mu}_k)(\boldsymbol{x}_v\boldsymbol{\mu}_k)^2\iota} + 3\ell_j\iota}{\Phi_k^j(\boldsymbol{\mu}_k)}. \tag{26}$$

We defer the proof of the claim to Appendix D.4 and continue to bound the three terms in (26). For the second term, we have

$$\sum_{j=1}^{L_2} \sum_{k \in \mathcal{K}_j} \boldsymbol{x}_k \boldsymbol{\mu}_k \frac{\sqrt{\sum_{v=1}^{k-1} 2\mathsf{clip}_j^2(\boldsymbol{x}_v\boldsymbol{\mu}_k)(\boldsymbol{x}_v\boldsymbol{\mu}_k)^2\iota}}{\Phi_k^j(\boldsymbol{\mu}_k)}$$

$$\leqslant \frac{1}{2} \sum_{j=1}^{L_2} \sum_{k \in \mathcal{K}_j} \boldsymbol{x}_k \boldsymbol{\mu}_k + \sum_{j=1}^{L_2} \sum_{k \in \mathcal{K}_j} \boldsymbol{x}_k \boldsymbol{\mu}_k \mathbb{I}\left\{ \frac{\sqrt{\sum_{v=1}^{k-1} 2\mathsf{clip}_j^2(\boldsymbol{x}_v\boldsymbol{\mu}_k)(\boldsymbol{x}_v\boldsymbol{\mu}_k)^2\iota}}{\Phi_k^j(\boldsymbol{\mu}_k)} > \frac{1}{2} \right\}. \tag{27}$$

We note that

$$\sum_{j=1}^{L_2} \sum_{k \in \mathcal{K}_j} \boldsymbol{x}_k \boldsymbol{\mu}_k \mathbb{I}\left\{ \frac{\sqrt{\sum_{v=1}^{k-1} 2\mathsf{clip}_j^2(\boldsymbol{x}_v\boldsymbol{\mu}_k)(\boldsymbol{x}_v\boldsymbol{\mu}_k)^2\iota}}{\Phi_k^j(\boldsymbol{\mu}_k)} > \frac{1}{2} \right\} \leqslant \sum_{j=1}^{L_2} \sum_{k \in \mathcal{K}_j} \boldsymbol{x}_k \boldsymbol{\mu}_k \mathbb{I}\left\{ \Phi_k^j(\boldsymbol{\mu}_k) \leqslant 4\ell_j\iota \right\}$$

$$\leqslant \sum_{j=1}^{L_2} \sum_{k \in \mathcal{K}_j} \boldsymbol{x}_k \boldsymbol{\mu}_k \frac{4\ell_j\iota}{\Phi_k^j(\boldsymbol{\mu}_k)}$$

$$\leqslant \sum_{j=1}^{L_2} \sum_{k \in \mathcal{K}_j} \frac{4\mathsf{clip}_j^2(\boldsymbol{x}_k\boldsymbol{\mu}_k)\iota}{\Phi_k^j(\boldsymbol{\mu}_k)}$$

$$\leqslant O(d^4|\Lambda_2|\iota \log^3(dK)), \tag{28}$$

where the last inequality uses Lemma 17. Collecting (26),(27) and (28), we have

$$\sum_k \boldsymbol{x}_k \boldsymbol{\mu}_k \leqslant 1 + \sum_{j=1}^{L_2} \sum_{k \in \mathcal{K}_j} 3\boldsymbol{x}_k \boldsymbol{\mu}_k \times \frac{\sqrt{\Psi_k^j(\boldsymbol{\mu}_k)\iota} + \ell_j\iota}{\Phi_k^j(\boldsymbol{\mu}_k)} + \frac{1}{2} \sum_{j=1}^{L_2} \sum_{k \in \mathcal{K}_j} \boldsymbol{x}_k \boldsymbol{\mu}_k + O(d^4|\Lambda_2|\iota \log^3(dK)).$$

Solving $\sum_k \boldsymbol{x}_k \boldsymbol{\mu}_k$, we obtain

$$\sum_k \boldsymbol{x}_k \boldsymbol{\mu}_k \leqslant O(d^4|\Lambda_2|\iota \log^3(dK)) + \sum_{j=1}^{L_2} \sum_{k \in \mathcal{K}_j} 6\boldsymbol{x}_k \boldsymbol{\mu}_k \times \frac{\sqrt{\Psi_k^j(\boldsymbol{\mu}_k)\iota} + \ell_j\iota}{\Phi_k^j(\boldsymbol{\mu}_k)}$$

$$\leqslant O(d^4|\Lambda_2|\iota \log^3(dK)) + \sum_{j=1}^{L_2} \sum_{k \in \mathcal{K}_j} 12\boldsymbol{x}_k \boldsymbol{\mu}_k \times \frac{\sqrt{\Psi_k^j(\boldsymbol{\mu}_k)\iota}}{\Phi_k^j(\boldsymbol{\mu}_k)}, \tag{29}$$

where (29) uses the last two steps in (28). The remaining term in (29) can be bounded as

$$\sum_{j=1}^{L_2}\sum_{k\in\mathcal{K}_j} 12\boldsymbol{x}_k\boldsymbol{\mu}_k \times \frac{\sqrt{\Psi_k^j(\boldsymbol{\mu}_k)\iota}}{\Phi_k^j(\boldsymbol{\mu}_k)} \leqslant \sum_{j=1}^{L_2}\sum_{k\in\mathcal{K}_j} 12\boldsymbol{x}_k\boldsymbol{\mu}_k\ell_j \frac{\sqrt{\sum_{v=1}^{k-1}\eta_v(\boldsymbol{\theta}^*)\iota}}{\Phi_k^j(\boldsymbol{\mu}_k)} \tag{30}$$

$$\leqslant \sum_{j=1}^{L_2}\sum_{k\in\mathcal{K}_j} \frac{12\boldsymbol{x}_k\boldsymbol{\mu}_k\ell_j}{\Phi_k^j(\boldsymbol{\mu}_k)}\sqrt{\sum_{k=1}^{K}\eta_k(\boldsymbol{\theta}^*)\iota}$$

$$\leqslant O(d^4|\Lambda_2|\log^3(dK)) \times \sqrt{\sum_{k=1}^{K}\eta_k(\boldsymbol{\theta}^*)\iota} \tag{31}$$

$$\leqslant O(d^4|\Lambda_2|\log^3(dK)) \times \sqrt{\left(\ln\frac{1}{\delta}+\sum_{k=1}^{K}\sigma_k^2\right)\iota}, \tag{32}$$

where (30) uses the definition of $\Psi_k^j(\cdot)$, (31) again uses the last two steps in (28), and (32) uses the event $\mathcal{E}_2$. $\qquad\square$

Now we can finish the proof of Theorem 3. We choose $\delta = O((K\log K)^{-1})$. Since on the event $\mathcal{E}^C$, we have $\mathfrak{R}^K \leqslant K$. Therefore, together with the bound on $\mathcal{E}$ from Lemma 19, we conclude that the expected regret is bounded by $\mathbb{E}[\mathfrak{R}^K] \leqslant \widetilde{O}(d^{4.5}\sqrt{\sum_{k=1}^{K}\sigma_k^2} + d^5)$.

*Proof.* It suffices to prove the theorem for $b = 1$, because otherwise we can apply $\{X_i/b\}_{i=1}^n$ to the $b = 1$ case. By Lemma 11 with $\epsilon = 1$ and $\delta < 1/e$, we have

$$\Pr\left[\left|\sum_{i=1}^{n}X_i\right| \geqslant 2\sqrt{\sum_{i=1}^{n}2\,\mathbb{E}[X_i^2\mid\mathcal{F}_{i-1}]\ln\frac{1}{\delta}}+4\ln\frac{1}{\delta}\right] \leqslant 4\delta\log_2 n. \tag{33}$$

By Lemma 9, we have

$$\Pr\left[\sum_{i=1}^{n}\mathbb{E}[X_i^2\mid\mathcal{F}_{i-1}] \geqslant \sum_{i=1}^{n}8X_i^2 + 4\ln\frac{4}{\delta}\right] \leqslant (\lceil\log_2 n\rceil+1)\delta. \tag{34}$$

Therefore, by a union bound over (33) and (34), we have with probability at least $1 - 6\delta\log_2 n$,

$$\left|\sum_{i=1}^{n}X_i\right| \leqslant \sqrt{\sum_{i=1}^{n}8\,\mathbb{E}[X_i^2\mid\mathcal{F}_{i-1}]\ln\frac{1}{\delta}}+4\ln\frac{1}{\delta}$$

$$\leqslant \sqrt{8\left(\sum_{i=1}^{n}8X_i^2 + 4\ln\frac{4}{\delta}\right)\ln\frac{1}{\delta}}+4\ln\frac{1}{\delta} \leqslant 8\sqrt{\sum_{i=1}^{n}X_i^2\ln\frac{1}{\delta}}+16\ln\frac{1}{\delta},$$

which concludes the proof. $\qquad\square$

### D.3    Proof of Lemma 18

*Proof.* Let $\delta' = e^{-\iota}$. We define the desired event $\mathcal{E} = \bigcap_{k\in[K],j\in\Lambda_2}\mathcal{E}_k^j$, where

$$\mathcal{E}_k^j = \left\{\left|\sum_{v=1}^{k}\mathsf{clip}_j(\boldsymbol{x}_v\boldsymbol{\mu})\epsilon_v(\boldsymbol{\theta}^*)\right| \leqslant \sqrt{\sum_{v=1}^{k}\mathsf{clip}_j^2(\boldsymbol{x}_v\boldsymbol{\mu})\eta_v(\boldsymbol{\theta}^*)\iota}+\ell_j\iota, \forall\boldsymbol{\mu}\in\mathcal{B}\right\}.$$

Note that for each $v$, we have that $\left|\text{clip}_j(\boldsymbol{x}_v\boldsymbol{\mu})\epsilon_v(\boldsymbol{\theta}^*)\right| \leqslant \ell_j$ and that $(\text{clip}_j(\boldsymbol{x}_v\boldsymbol{\mu})\epsilon_v(\boldsymbol{\theta}^*))^2 = \text{clip}_j^2(\boldsymbol{x}_v\boldsymbol{\mu})\eta_v(\boldsymbol{\theta}^*)$, so by Theorem 4, we have

$$\Pr\left[\left|\sum_{v=1}^k \text{clip}_j(\boldsymbol{x}_v\boldsymbol{\mu})\varepsilon_v\right| \leqslant \sqrt{\sum_{v=1}^k \text{clip}_j^2(\boldsymbol{x}_v\boldsymbol{\mu})\text{Var}(\varepsilon_v \mid \mathcal{F}_v)\iota + \ell_j\iota}\right]$$

$$\geqslant 1 - O\left(e^{-\frac{\iota}{\log_2\log_2 K}}\right)$$

$$\geqslant 1 - O\left(\frac{\delta}{K|\mathcal{B}||\Lambda_2|}\log K\right),$$

where $\mathcal{F}_v$ is as defined in Section 3. Finally, using a union bound over $(\boldsymbol{\mu}, j, k) \in \mathcal{B} \times \Lambda_2 \times [K]$, we have $\Pr[\mathcal{E}] \geqslant 1 - O(\delta \log K)$. $\qquad\square$

## D.4   Proof of Claim 20

*Proof.* We elaborate on (26). We will prove it by showing that the numerator is always greater than the denominator in the fraction in (26), so each term $\boldsymbol{x}_k\boldsymbol{\mu}_k$ is multiplied by a number greater than 1. We have for every $j \in \Lambda_2$,

$$\Phi_k^j(\boldsymbol{\mu}_k) = \sum_{v=1}^{k-1} \text{clip}_j(\boldsymbol{x}_v\boldsymbol{\mu}_k)\boldsymbol{x}_v\boldsymbol{\mu}_k + \ell_j^2$$

$$\leqslant \left|\sum_{v=1}^{k-1} \text{clip}_j(\boldsymbol{x}_v\boldsymbol{\mu}_k)\epsilon_v(\boldsymbol{\theta}^*)\right| + \left|\sum_{v=1}^{k-1} \text{clip}_j(\boldsymbol{x}_v\boldsymbol{\mu}_k)\epsilon_v(\boldsymbol{\theta}_k)\right| + \ell_j^2 \tag{35}$$

$$\leqslant \sqrt{\Psi_k^j(\boldsymbol{\mu}_k)\iota} + \sqrt{\sum_{v=1}^{k-1} \text{clip}_j^2(\boldsymbol{x}_v\boldsymbol{\mu}_k)\eta_v(\boldsymbol{\theta}_k)\iota + l_j^2 + \frac{1}{HK}} + 2\ell_j\iota \tag{36}$$

$$\leqslant \sqrt{\Psi_k^j(\boldsymbol{\mu}_k)\iota} + \sqrt{\sum_{v=1}^{k-1} \text{clip}_j^2(\boldsymbol{x}_v\boldsymbol{\mu}_k)\eta_v(\boldsymbol{\theta}^*)\iota} + \sqrt{\sum_{v=1}^{k-1} \text{clip}_j^2(\boldsymbol{x}_v\boldsymbol{\mu}_k)|\eta_v(\boldsymbol{\theta}_k) - \eta_v(\boldsymbol{\theta}^*)|\iota} + 3\ell_j\iota$$

$$= 2\sqrt{\Psi_k^j(\boldsymbol{\mu}_k)\iota} + \sqrt{\sum_{v=1}^{k-1} \text{clip}_j^2(\boldsymbol{x}_v\boldsymbol{\mu}_k)|\eta_v(\boldsymbol{\theta}_k) - \eta_v(\boldsymbol{\theta}^*)|\iota} + 3\ell_j\iota$$

$$\leqslant 2\sqrt{\Psi_k^j(\boldsymbol{\mu}_k)\iota} + \sqrt{\sum_{v=1}^{k-1} \text{clip}_j^2(\boldsymbol{x}_v\boldsymbol{\mu}_k)(\eta_v(\boldsymbol{\theta}^*) + 2(\boldsymbol{x}_v\boldsymbol{\mu}_k)^2)\iota} + 3\ell_j\iota \tag{37}$$

$$\leqslant 2\sqrt{\Psi_k^j(\boldsymbol{\mu}_k)\iota} + \sqrt{\sum_{v=1}^{k-1} \text{clip}_j^2(\boldsymbol{x}_v\boldsymbol{\mu}_k)\eta_v(\boldsymbol{\theta}^*)\iota} + \sqrt{\sum_{v=1}^{k-1} 2\text{clip}_j^2(\boldsymbol{x}_v\boldsymbol{\mu}_k)(\boldsymbol{x}_v\boldsymbol{\mu}_k)^2\iota} + 3\ell_j\iota,$$

$$= 3\sqrt{\Psi_k^j(\boldsymbol{\mu}_k)\iota} + \sqrt{\sum_{v=1}^{k-1} 2\text{clip}_j^2(\boldsymbol{x}_v\boldsymbol{\mu}_k)(\boldsymbol{x}_v\boldsymbol{\mu}_k)^2\iota} + 3\ell_j\iota, \tag{38}$$

where (35) uses $\epsilon_v(\boldsymbol{\theta}_k) - \epsilon_v(\boldsymbol{\theta}^*) = \boldsymbol{x}_v(\boldsymbol{\theta}_k - \boldsymbol{\theta}^*) = \boldsymbol{x}_v\boldsymbol{\mu}_k$, (36) uses that $\boldsymbol{\theta}^*, \boldsymbol{\theta}_k \in \Theta_k$, the definition of $\Theta_k$ in (1) and the fact that $\mathcal{B}$ is an $K^{-3}$-net of $\mathbb{B}_2^d(2)$, and (37) uses

$$|\eta_v(\boldsymbol{\theta}_k) - \eta_v(\boldsymbol{\theta}^*)| = \left|(\epsilon_v(\boldsymbol{\theta}^*) - \boldsymbol{x}_v\boldsymbol{\mu}_k)^2 - (\epsilon_v(\boldsymbol{\theta}^*))^2\right|$$

$$\leqslant 2|\epsilon_v(\boldsymbol{\theta}^*)|\boldsymbol{x}_v\boldsymbol{\mu}_k + (\epsilon_v(\boldsymbol{\theta}^*))^2 \leqslant (\boldsymbol{x}_v\boldsymbol{\mu}_k)^2 + 2(\epsilon_v(\boldsymbol{\theta}^*))^2.$$

Since (38) holds for every $j \in \Lambda_2$, it holds for $j = j_k$, and thus (26) follows. $\qquad\square$

# E  Missing Proofs in Section 5

## E.1  Proof of Theorem 5

Before introducing our proof, we make some definitions. We let $\boldsymbol{\theta}_{k,h}^m = \arg\max_{\boldsymbol{\theta}\in\Theta_k} |\boldsymbol{x}_{k,h}^m(\boldsymbol{\theta}-\boldsymbol{\theta}^*)|$ and $\boldsymbol{\mu}_{k,h}^m = \boldsymbol{\theta}_{k,h}^m - \boldsymbol{\theta}^*$. Recall that $\mathcal{T}_k^{m,i}$ is defined in Algorithm 2. We define

$$\Phi_k^{m,i,j}(\boldsymbol{\mu}) = \sum_{(v,u)\in\mathcal{T}_k^{m,i}} \mathsf{clip}_j(\boldsymbol{x}_{v,u}^m\boldsymbol{\mu})\boldsymbol{x}_{v,u}^m\boldsymbol{\mu} + \ell_j^2, \quad \Psi_k^{m,i,j}(\boldsymbol{\mu}) = \sum_{(v,u)\in\mathcal{T}_k^{m,i}} \mathsf{clip}_j^2(\boldsymbol{x}_{v,u}^m\boldsymbol{\mu})\eta_{v,u}^m. \quad (39)$$

Note that our definitions in (39) are similar to those for linear bandits in (25). The main differences are: 1) we define $\Phi(\cdot), \Psi(\cdot)$ also for higher moments, as indicated by the index $m$ in their superscripts; 2) we add the variance layer, so that we only use samples from $\mathcal{T}^{m,i}$; 3) since we can now estimate variance, we use the upper bound of estimated variance in lieu of the empirical variance. For $h \in [H+1]$, we further define

$$\mathcal{T}_{k,h}^{m,i} = \left\{(v,u)\in[k-1]\times[H] \cup \{(k,u):u<h\}:\eta_{v,u}^m\in(l_{i+1},l_i]\right\}, \forall 1\leqslant i\leqslant L_1;$$

$$\mathcal{T}_{k,h}^{m,L_1+1} = \left\{(v,u)\in[k-1]\times[H]\cup\{(k,u):u<h\}:\eta_{v,u}^m\leqslant l_{L_1+1}\right\};$$

$$\Phi_{k,h}^{m,i,j}(\boldsymbol{\mu}) = \sum_{(v,u)\in\mathcal{T}_{k,h}^{m,i}} \mathsf{clip}_j(\boldsymbol{x}_{v,u}^m\boldsymbol{\mu})\boldsymbol{x}_{v,u}^m\boldsymbol{\mu} + \ell_j^2, \quad \Psi_{k,h}^{m,i,j}(\boldsymbol{\mu}) = \sum_{(v,u)\in\mathcal{T}_{k,h}^{m,i}} \mathsf{clip}_j^2(\boldsymbol{x}_{v,u}^m\boldsymbol{\mu})\eta_{v,u}^m; \quad (40)$$

$$I_h^k = \mathbb{I}\{\forall u\leqslant h, m, i, j: \Phi_{k,u}^{m,i,j}(\boldsymbol{\mu}_{k,u}^m)\leqslant 4(d+2)^2\Phi_k^{m,i,j}(\boldsymbol{\mu}_{k,u}^m)\}.$$

Here $I_h^k = 1$ indicates that for every $u\leqslant h$, the confidence set using data prior to the time step $(k,u)$ can be properly approximated by the confidence set with data prior to the episode $k$. We define $I_h^k$ in this way to ensure that it is $\mathcal{F}_h^k$-measurable. The following lemma ensures that $Q_h^k$ is optimistic with high probability. Its proof is deferred to Appendix E.3.

**Lemma 21.** $\Pr\left[\forall k, h, s, a: Q_h^k(s,a)\geqslant Q_h^*(s,a)\right]\geqslant\Pr[\forall k\in[K]:\boldsymbol{\theta}^*\in\Theta_k]\geqslant 1-O(\delta)$.

When the event specified in Lemma 21 holds, the regret can be decomposed as

$$\mathfrak{R}^K = \sum_{k=1}^K \left(V_1^*(s_1^k)-V_1^{\pi_k}(s_1^k)\right)\leqslant\sum_{k=1}^K\left(V_1^k(s_1^k)-V_1^{\pi_k}(s_1^k)\right)\leqslant\check{\mathfrak{R}}_1+\check{\mathfrak{R}}_2+\mathfrak{R}_3+\sum_{k,h}(I_h^k-I_{h+1}^k),$$

where

$$\check{\mathfrak{R}}_1 = \sum_{k,h}(P_{s_h^k,a_h^k}V_{h+1}^k-V_{h+1}^k(s_{h+1}^k))I_h^k, \qquad \check{\mathfrak{R}}_2 = \sum_{k,h}\left(V_h^k(s_h^k)-r_h^k-P_{s_h^k,a_h^k}V_{h+1}^k\right)I_h^k,$$

$$\mathfrak{R}_3 = \sum_{k=1}^K\left(\sum_{h=1}^H r_h^k - V_1^{\pi_k}(s_1^k)\right).$$

Next we analyze these terms. First, we observe that $\mathfrak{R}_3$ is a sum of a martingale difference sequence, so by Lemma 6, we have $\mathfrak{R}_3\leqslant O(\sqrt{K\log(1/\delta)})$ with probability at least $1-\delta$. Next, we use the following lemma to bound $\sum_{k,h}(I_h^k-I_{h+1}^k)$. We defer its proof to Appendix E.4.

**Lemma 22.** $\sum_{k,h}(I_h^k-I_{h+1}^k)\leqslant O(d\log^5(dHK))$.

To bound $\check{\mathfrak{R}}_1$ and $\check{\mathfrak{R}}_2$, we need to define the following quantities. First, we denote $\check{\boldsymbol{x}}_{k,h} = \boldsymbol{x}_{k,h}I_h^k$ and $\check{\eta}_{k,h}^m = \eta_{k,h}^mI_h^k$. Next, for $m\in\Lambda_0$, we define

$$\check{R}_m = \sum_{k,h}\left|\check{\boldsymbol{x}}_{k,h}^m\boldsymbol{\mu}_{k,h}^m\right|, \qquad \check{M}_m = \sum_{k,h}\left(P_{s_h^k,a_h^k}(V_{h+1}^k)^{2^m} - (V_{h+1}^k(s_{h+1}^k))^{2^m}\right)I_h^k.$$

Intuitively, $\check{R}_m$ represents the "regret" of $2^m$-th moment prediction and $\check{M}_m$ represents the total variance of $2^m$-th order value function. We have $\check{\mathfrak{R}}_1 = \check{M}_0$ by definition and and using that

$$Q_h^k(s,a)-r(s,a)-P_{s,a}V_{h+1}^k\leqslant\max_{\boldsymbol{\theta}\in\Theta_k}\boldsymbol{x}_{k,h}^0(\boldsymbol{\theta}-\boldsymbol{\theta}^*),$$

we have $\check{\mathfrak{R}}_2\leqslant\check{R}_0$. So it suffices to bound $\check{R}_0+\check{M}_0$, which is done by the following lemma.

**Lemma 23.** *With probability at least $1 - \delta$, we have*

$$\check{R}_0 + \left|\check{M}_0\right| \leqslant O\left(d^{4.5}\sqrt{K\log^5(dHK)\log(1/\delta)} + d^9\log^6(dHK)\log(1/\delta)\right).$$

Lemma 23 is the main technical part of our result in Section 5, so we sketch its proof in the next subsection. With the lemma in hand, we have with probability $1 - \delta$ that $\mathfrak{R}^K \leqslant \widetilde{O}(d^{4.5}\sqrt{K} + d^9)$. Finally, We conclude the proof to Theorem 5 by choosing $\delta = 1/K$ and noting that $\mathfrak{R}^K \leqslant K$.

## E.2 Bounding $\check{R}$ and $\check{M}$

We sketch the proof for Lemma 23. The first step to bound $\check{R}_m$ is to relate it to the variance $\check{\eta}^m$.

**Lemma 24.** *With probability at least $1 - \delta$, we have $\check{R}_m \leqslant O(d^4\sqrt{\sum_{k,h}\check{\eta}_{k,h}^m \iota \log^7(dHK)} + d^6 \iota \log^5(dHK))$.*

We defer the proof to Appendix E.5. The proof is spiritually similar to proof of Lemma 19. The main difference is that we use the peeling technique to the magnitude of the variance.

Based on Lemma 24, we use the following recursion lemma to relate $\check{R}_m, \check{M}_m$ to $\check{R}_{m+1}, \check{M}_{m+1}$. We defer the proof to Appendix E.6. It mainly uses similar ideas in Zhang et al. [2020a].

**Lemma 25** (Recursions)**.** *With probability at least $1 - \delta$, we have*

$$\check{R}_m \leqslant O\left(d^4\sqrt{(\check{M}_{m+1} + 2^{m+1}(K + \check{R}_0) + \check{R}_{m+1} + \check{R}_m)\iota \log^7(dHK)} + d^6 \iota \log^5(dHK)\right),$$

$$\left|\check{M}_m\right| \leqslant O\left(\sqrt{(\check{M}_{m+1} + O(d\log^5(dHK)) + 2^{m+1}(K + \check{R}_0))\log(1/\delta)} + \log(1/\delta)\right).$$

Finally, we can prove Lemma 23 by collecting Lemma 24,25 and using a technical lemma about recursion (Lemma 12). The details are in Appendix E.7.

## E.3 Proof of Lemma 21

*Proof.* The lemma consists of two inequalities. The first inequality is proved using backward induction, where the induction step is given as

$$Q_h^k(s,a) = \min\{1, r(s,a) + \max_{\boldsymbol{\theta}\in\Theta_k}\sum_{i=1}^d \theta_i P_{s,a}^i V_{h+1}^k\}$$

$$\geqslant \min\{1, r(s,a) + \sum_{i=1}^d \boldsymbol{\theta}_i^* P_{s,a}^i V_{h+1}^k\} \geqslant \min\{1, r(s,a) + \sum_{i=1}^d \theta_i^* P_{s,a}^i V_{h+1}^*\} = Q_h^*(s,a),$$

$$V_h^k(s) = \max_a Q_h^k(s,a) \geqslant \max_a Q_h^*(s,a) = V_h^*(s).$$

We now prove the second inequality. Let $\delta' = e^{-\iota}$. We define the desired event $\mathcal{E} = \bigcap_{k,m,i,j}\mathcal{E}_k^{m,i,j}$, where

$$\mathcal{E}_k^{m,i,j} = \left\{\left|\sum_{(v,u)\in\mathcal{T}_k^{m,i}}\mathsf{clip}_j(\boldsymbol{x}_{v,u}^m\boldsymbol{\mu})\varepsilon_{\kappa,h}^m\right| \leqslant 4\sqrt{\sum_{(v,u)\in\mathcal{T}_k^{m,i}}\mathsf{clip}_j^2(\boldsymbol{x}_{v,u}^m\boldsymbol{\mu})\mathrm{Var}(\varepsilon_{v,u}^m \mid \mathcal{F}_u^v)\ln\frac{1}{\delta'}} + 4\ell_j\ln\frac{1}{\delta'}, \forall\boldsymbol{\mu}\in\mathcal{B}\right\}.$$

Note that for a fixed $k$, we have that $|\mathsf{clip}_j(\boldsymbol{x}_{v,u}^m\boldsymbol{\mu})\varepsilon_{v,u}^m| \leqslant \ell_j \leqslant 1$ and that

$$\mathrm{Var}\left(\mathsf{clip}_j(\boldsymbol{x}_{v,u}^m\boldsymbol{\mu})\varepsilon_{v,u}^m\mathbb{I}\{(v,u)\in\mathcal{T}_k^{m,i}\} \mid \mathcal{F}_u^v\right) = \mathsf{clip}_j(\boldsymbol{x}_{k,h}^m\boldsymbol{\mu})^2\mathbb{I}\{(v,u)\in\mathcal{T}_k^{m,i}\}\mathrm{Var}(\varepsilon_{v,u}^m \mid \mathcal{F}_u^v),$$

so by Lemma 11 with $b = \ell_j, \epsilon = 1$, we have

$$\Pr\left[\left|\sum_{(v,u)\in\mathcal{T}_k^{m,i}}\mathsf{clip}_j(\boldsymbol{x}_{v,u}^m\boldsymbol{\mu})\varepsilon_{v,u}^m\right| \geqslant 4\sqrt{\sum_{(v,u)\in\mathcal{T}_k^{m,i}}\mathsf{clip}_j^2(\boldsymbol{x}_{v,u}^m\boldsymbol{\mu})\mathrm{Var}(\varepsilon_{v,u}^m \mid \mathcal{F}_u^v)\ln\frac{1}{\delta'}} + 4\ell_j\ln\frac{1}{\delta'}\right]$$

$$\leqslant 4\delta'\log_2(HK).$$

Using a union bound over $(\boldsymbol{\mu}, m, i, j, k) \in \mathcal{B} \times \Lambda_0 \times \Lambda_1 \times \Lambda_2 \times [K]$, we have $\Pr[\mathcal{E}] \geqslant 1 - O(\delta' K |\mathcal{B}| \log^4(HK)) \geqslant 1 - O(\delta)$.

Next we show that the event $\mathcal{E}$ implies that $\boldsymbol{\theta}^* \in \Theta_k$ for every $k \in [K]$. We show by induction over $k$. For $k = 1$ it is clear. For $k \geqslant 1$, since $\boldsymbol{\theta}^* \in \Theta_k$, for every $h \in [H]$, we have $\eta_{k,h}^m = \max_{\boldsymbol{\theta} \in \Theta_k} \{\boldsymbol{\theta} \boldsymbol{x}_{k,h}^{m+1} - (\boldsymbol{\theta} \boldsymbol{x}_{k,h}^m)^2\} \geqslant \boldsymbol{\theta}^* \boldsymbol{x}_{k,h}^{m+1} - (\boldsymbol{\theta}^* \boldsymbol{x}_{k,h}^m)^2 \geqslant \mathrm{Var}(\varepsilon_{k,h}^m \mid \mathcal{F}_h^k)$, which, together with the event $\bigcap_{m,i,j} \mathcal{E}_{k+1}^{m,i,j}$, implies that $\boldsymbol{\theta}^* \in \Theta_{k+1}$. $\square$

### E.4 Proof of Lemma 22

*Proof.* We define
$$I_{k,h}^{m,i,j} = \mathbb{I}\{\forall u \leqslant h : \Phi_{k,u}^{m,i,j}(\boldsymbol{\mu}_{k,u}^m) \leqslant 4(d+2)^2 \Phi_k^{m,i,j}(\boldsymbol{\mu}_{k,u}^m)\}.$$

Then we have $I_h^k = \prod_{m,i,j} I_{k,h}^{m,i,j}$. Also we have
$$\sum_h (I_h^k - I_{h+1}^k) \leqslant \sum_{m,i,j} \sum_h (I_{k,h}^{m,i,j} - I_{k,h+1}^{m,i,j}).$$

Note that $I_h^k \geqslant I_{h+1}^k$ and $I_{k,h}^{m,i,j} \geqslant I_{k,h+1}^{m,i,j}$. For each fixed $m, i, j$, if $\sum_h (I_{k,h}^{m,i,j} - I_{k,h+1}^{m,i,j}) = 1$, then there exists $h \in [H]$, such that for the time step $(k, h)$, we have $\Phi_{k,h}^{m,i,j}(\boldsymbol{\mu}) > 4(d+2)^2 \Phi_k^{m,i,j}(\boldsymbol{\mu})$ for some $\boldsymbol{\mu}$. By Lemma 14 with $f(x) = \mathrm{clip}(x, \ell_j)x$ and $\ell = \ell_j$, there are at most $O(d \log^2(dHK))$ such time steps. We conclude by noting that we have $|\Lambda_0 \times \Lambda_1 \times \Lambda_2| \leqslant O(\log^3(dHK))$ possible $m, i, j$ pairs. $\square$

### E.5 Proof of Lemma 24

To prove this lemma, we define the index sets to help us apply the peeling technique. We denote
$$\mathcal{T}_k^{m,i,j} = \{(v, u) \in \mathcal{T}_k^{m,i} : |\boldsymbol{x}_{v,u}^m \boldsymbol{\mu}_{v,u}^m| \in (\ell_{j+1}, \ell_j]\},$$
$$\mathcal{T}_k^{m,i,L_2+1} = \{(v, u) \in \mathcal{T}_k^{m,i} : |\boldsymbol{x}_{v,u}^m \boldsymbol{\mu}_{v,u}^m| \in [0, \ell_{L_2+1}]\},$$
and $\check{\mathcal{T}}_k^{m,i,j} = \{(v, u) \in \mathcal{T}_k^{m,i,j} : I_u^v = 1\}$. We also denote $\mathcal{T}^{m,i,j} = \mathcal{T}_{K+1}^{m,i,j}, \check{\mathcal{T}}^{m,i,j} = \check{\mathcal{T}}_{K+1}^{m,i,j}$.

*Proof.* Since $\boldsymbol{\theta}_{k,h}^m \in \Theta_k \subseteq \Theta_k^{m,i,j}$, choosing $\boldsymbol{\mu} = \boldsymbol{\mu}_{k,h}^m$ in the confidence set definition and using that $\boldsymbol{x}_{v,u}^m \boldsymbol{\mu}_{k,h}^m = \epsilon_{v,u}^m(\boldsymbol{\theta}^*) - \epsilon_{v,u}^m(\boldsymbol{\theta}_{k,h}^m)$, we have

$$\Phi_k^{m,i,j}(\boldsymbol{\mu}_{k,h}^m) = \sum_{(v,u) \in \mathcal{T}_k^{m,i}} \mathrm{clip}_j(\boldsymbol{x}_{v,u}^m \boldsymbol{\mu}_{k,h}^m) \boldsymbol{x}_{v,u}^m \boldsymbol{\mu}_{k,h}^m + \ell_j^2$$

$$\leqslant \left| \sum_{(v,u) \in \mathcal{T}_k^{m,i}} \mathrm{clip}_j(\boldsymbol{x}_{v,u}^m \boldsymbol{\mu}_{k,h}^m) \epsilon_{v,u}^m(\boldsymbol{\theta}^*) \right| + \left| \sum_{(v,u) \in \mathcal{T}_k^{m,i}} \mathrm{clip}_j(\boldsymbol{x}_{v,u}^m \boldsymbol{\mu}_{k,h}^m) \epsilon_{v,u}^m(\boldsymbol{\theta}_{k,h}^m) \right| + \ell_j^2$$

$$\leqslant 8 \sqrt{\sum_{(v,u) \in \mathcal{T}_k^{m,i}} \mathrm{clip}_j(\boldsymbol{x}_{v,u}^m \boldsymbol{\mu}_{k,h}^m) \eta_{v,u}^m \iota} + 8\ell_j \iota + \ell_j^2$$

$$\leqslant 8 \sqrt{\Psi_k^{m,i,j}(\boldsymbol{\mu}_{k,h}^m) \iota} + 16\ell_j \iota. \tag{41}$$

Therefore, when $I_k^h = 0$, we have
$$\frac{\Phi_{k,h}^{m,i,j}(\boldsymbol{\mu}_{k,h}^m)}{4(d+2)^2} \leqslant \Phi_k^{m,i,j}(\boldsymbol{\mu}_{k,h}^m) \leqslant 16 \left( \sqrt{\Psi_{k,h}^{m,i,j}(\boldsymbol{\mu}_{k,h}^m) \iota} + \ell_j \iota \right).$$

Next we analyze the sum. Using the fact that
$$\frac{64(d+2)^2 \left( \sqrt{\Psi_{k,h}^{m,i,j}(\boldsymbol{\mu}_{k,h}^m) \iota} + \ell_j \iota \right)}{\Phi_{k,h}^{m,i,j}(\boldsymbol{\mu}_{k,h}^m)} \geqslant 1,$$

we obtain

$$\sum_{(k,h)\in\check{\mathcal{T}}^{m,i,j}} \left|\boldsymbol{x}_{k,h}^m\boldsymbol{\mu}_{k,h}^m\right| \leqslant \sum_{(k,h)\in\check{\mathcal{T}}^{m,i,j}} \left|\boldsymbol{x}_{k,h}^m\boldsymbol{\mu}_{k,h}^m\right| \frac{64(d+2)^2\left(\sqrt{\Psi_{k,h}^{m,i,j}(\boldsymbol{\mu}_{k,h}^m)\iota} + \ell_j\iota\right)}{\Phi_{k,h}^{m,i,j}(\boldsymbol{\mu}_{k,h}^m)} \tag{42}$$

$$\leqslant 64(d+2)^2 \sum_{(k,h)\in\check{\mathcal{T}}^{m,i,j}} \left(\frac{\left|\boldsymbol{x}_{k,h}^m\boldsymbol{\mu}_{k,h}^m\right|\sqrt{\ell_i\iota}}{\sqrt{\Phi_{k,h}^{m,i,j}(\boldsymbol{\mu}_{k,h}^m)}} + \frac{\left|\boldsymbol{x}_{k,h}^m\boldsymbol{\mu}_{k,h}^m\right|\ell_j\iota}{\Phi_{k,h}^{m,i,j}(\boldsymbol{\mu}_{k,h}^m)}\right), \tag{43}$$

where the last inequality uses that for every $\boldsymbol{\mu}$, we have

$$\Psi_{k,h}^{m,i,j}(\boldsymbol{\mu}) = \sum_{(v,u)\in\mathcal{T}_{k,h}^{m,i}} \mathsf{clip}_j^2(\boldsymbol{x}_{v,u}^m\boldsymbol{\mu})\eta_{v,u}^m \leqslant \ell_i \sum_{(v,u)\in\mathcal{T}_{k,h}^{m,i}} \mathsf{clip}_j(\boldsymbol{x}_{k,h}^m\boldsymbol{\mu})\boldsymbol{x}_{k,h}^m\boldsymbol{\mu} \leqslant \ell_i\Phi_{k,h}^{m,i,j}(\boldsymbol{\mu}). \tag{44}$$

In (44), the first inequality uses that $\eta_{v,u}^m \leqslant \ell_i$ for $(v,u)\in\mathcal{T}_{k,h}^{m,i}$ and that $\mathsf{clip}_j^2(\alpha) \leqslant \mathsf{clip}_j(\alpha)\alpha$ for $\alpha\in\mathbb{R}$, and the second inequality uses the definition of $\Phi_{k,h}^{m,i,j}(\boldsymbol{\mu})$. Next we bound the two terms in (43). To bound the first term, we note that

$$\sum_{(k,h)\in\check{\mathcal{T}}^{m,i,j}} \frac{\left|\boldsymbol{x}_{k,h}^m\boldsymbol{\mu}_{k,h}^m\right|}{\sqrt{\Phi_{k,h}^{m,i,j}(\boldsymbol{\mu}_{k,h}^m)}} \leqslant \sqrt{\left|\check{\mathcal{T}}^{m,i,j}\right|}\sqrt{\sum_{(k,h)\in\check{\mathcal{T}}^{m,i,j}} \frac{(\boldsymbol{x}_{k,h}^m\boldsymbol{\mu}_{k,h}^m)^2}{\Phi_{k,h}^{m,i,j}(\boldsymbol{\mu}_{k,h}^m)}} \tag{45}$$

$$\leqslant \sqrt{\left|\check{\mathcal{T}}^{m,i,j}\right|}\sqrt{\sum_{(k,h)\in\check{\mathcal{T}}^{m,i,j}} \frac{\mathsf{clip}_j^2(\boldsymbol{x}_{k,h}^m\boldsymbol{\mu}_{k,h}^m)}{\Phi_{k,h}^{m,i,j}(\boldsymbol{\mu}_{k,h}^m)}} \tag{46}$$

$$\leqslant \sqrt{\left|\check{\mathcal{T}}^{m,i,j}\right|}\sqrt{\sum_{(k,h)\in\check{\mathcal{T}}^{m,i,j}} \frac{\mathsf{clip}_j^2(\boldsymbol{x}_{k,h}^m\boldsymbol{\mu}_{k,h}^m)}{\sum\limits_{(v,u)\in\check{\mathcal{T}}_{k,h}^{m,i,j}} \mathsf{clip}_j(\boldsymbol{x}_{v,u}^m\boldsymbol{\mu}_{k,h}^m)\boldsymbol{x}_{v,u}^m\boldsymbol{\mu}_{k,h}^m + \ell_j^2}} \tag{47}$$

$$\leqslant \sqrt{\left|\check{\mathcal{T}}^{m,i,j}\right|} \times O(\sqrt{d^4\log^3(dHK)}), \tag{48}$$

where (45) uses Cauchy's inequality, (46) uses that $\left|\boldsymbol{x}_{k,h}^m\boldsymbol{\mu}_{k,h}^m\right| \leqslant \ell_j$ for $(k,h)\in\mathcal{T}^{m,i,j}$, (47) uses the definition of $\Phi_{k,h}^{m,i,j}(\boldsymbol{\mu})$, and (48) uses Lemma 17. To bound the second term in (43), we have

$$\sum_{(k,h)\in\check{\mathcal{T}}^{m,i,j}} \frac{\left|\boldsymbol{x}_{k,h}^m\boldsymbol{\mu}_{k,h}^m\right|\ell_j}{\Phi_{k,h}^{m,i,j}(\boldsymbol{\mu}_{k,h}^m)} \leqslant \sum_{(k,h)\in\mathcal{T}^{m,i,j}} \frac{2\mathsf{clip}_j^2(\boldsymbol{x}_{k,h}^m\boldsymbol{\mu}_{k,h}^m)}{\Phi_{k,h}^{m,i,j}(\boldsymbol{\mu}_{k,h}^m)} \leqslant O(d^4\log^3(dHK)), \tag{49}$$

where the first inequality uses that $\left|\boldsymbol{x}_{k,h}^m\boldsymbol{\mu}_{k,h}^m\right| \geqslant \ell_j/2$ for $(k,h)\in\mathcal{T}^{m,i,j}$ and the second inequality is the same as what we have shown from (46) to (48). As a result, combining (43),(48) and (49), we have

$$\sum_{(k,h)\in\check{\mathcal{T}}^{m,i,j}} \left|\boldsymbol{x}_{k,h}^m\boldsymbol{\mu}_{k,h}^m\right| \leqslant 64(d+2)^2 \times O\left(\sqrt{d^4\ell_i\left|\check{\mathcal{T}}^{m,i,j}\right|\iota\log^3(dHK)} + d^4\iota\log^3(dHK)\right) \tag{50}$$

$$\leqslant O\left(d^4\sqrt{\ell_i\left|\check{\mathcal{T}}^{m,i,j}\right|\iota\log^3(dHK)} + d^6\iota\log^3(dHK)\right). \tag{51}$$

Recall that (51) requires $\boldsymbol{x}_{k,h}^m\boldsymbol{\mu}_{k,h}^m \in [\ell_j/2, \ell_j]$, which would be false for $j = L_2 + 1$. In this corner case, $j = L_2 + 1$, we have

$$\sum_i \sum_{(k,h)\in\check{\mathcal{T}}^{m,i,j}} \left|\boldsymbol{x}_{k,h}^m\boldsymbol{\mu}_{k,h}^m\right| \leqslant KH\ell_j \leqslant O(1). \tag{52}$$

Finally, combining (51) and (52), we have

$$\sum_{k,h}\left|\check{\boldsymbol{x}}_{k,h}^m \boldsymbol{\mu}_{k,h}^m\right| = \sum_{i,j}\sum_{(k,h)\in\check{\mathcal{T}}^{m,i,j}}\left|\boldsymbol{x}_{k,h}^m \boldsymbol{\mu}_{k,h}^m\right|$$

$$\leqslant O(1) + \sum_{i,j}O\left(d^4\sqrt{\ell_i\left|\check{\mathcal{T}}^{m,i,j}\right|\iota\log^3(dHK)} + L_2 d^6\iota\log^3(dHK)\right)$$

$$\leqslant O\left(d^4\sqrt{\sum_{k,h}\check{\eta}_{k,h}^m\iota\log^7(dHK)} + d^6\iota\log^5(dHK)\right), \tag{53}$$

where (53) uses that $\ell_i\left|\check{\mathcal{T}}^{m,i,j}\right| \leqslant O(1 + \sum_{k,h}\check{\eta}_{k,h}^m)$, which can be proved as follows: for $i \leqslant L_1$, it is due to $\eta_{k,h}^m \geqslant \ell_i/2$; for $i = L_1 + 1$, it is due to $1/\ell_i \geqslant KH \geqslant \left|\check{\mathcal{T}}^{m,i,j}\right|$. $\qquad\square$

### E.6  Proof of Lemma 25

*Proof.* Define

$$\check{\zeta}_{k,h}^m = (P_{s_h^k,a_h^k}(V_{h+1}^k)^{2^{m+1}} - (P_{s_h^k,a_h^k}(V_{h+1}^k)^{2^m})^2)I_h^k.$$

We note that $\check{M}_m$ is a martingale, so by Lemma 11 with a union bound over $m$, we have

$$\Pr\left[\forall m \in \Lambda_0 : \left|\check{M}_m\right| \leqslant 2\sqrt{2\sum_{k,h}\check{\zeta}_{k,h}^m \ln\frac{1}{\delta}} + 4\ln\frac{1}{\delta}\right] \geqslant 1 - O(\delta\log^2(dKH)). \tag{54}$$

By the definition of $\check{\eta}_{k,h}^m$, we have

$$\sum_{k,h}\check{\eta}_{k,h}^m \leqslant \sum_{k,h}\left(\check{\zeta}_{k,h}^m + \max_{\boldsymbol{\theta}\in\Theta_k}\check{\boldsymbol{x}}_{k,h}^{m+1}(\boldsymbol{\theta} - \boldsymbol{\theta}^*) + 2\max_{\boldsymbol{\theta}\in\Theta_k}\check{\boldsymbol{x}}_{k,h}^m(\boldsymbol{\theta}^* - \boldsymbol{\theta})\right) \tag{55}$$

$$\leqslant \sum_{k,h}\check{\zeta}_{k,h}^m + \check{R}_{m+1} + 2\check{R}_m, \tag{56}$$

We have that

$$\sum_{k,h}\check{\zeta}_{k,h}^m = \sum_{k,h}\left(P_{s_h^k,a_h^k}(V_{h+1}^k)^{2^{m+1}} - (P_{s_h^k,a_h^k}(V_{h+1}^k)^{2^m})^2\right)I_h^k$$

$$\leqslant \sum_{k,h}\left(P_{s_h^k,a_h^k}(V_{h+1}^k)^{2^{m+1}} - (V_{h+1}^k(s_{h+1}^k))^{2^{m+1}}\right)I_h^k + \sum_{k,h}(V_h^k(s_h^k))^{2^{m+1}}(I_h^k - I_{h+1}^k)$$

$$+ \sum_{k,h}\left((V_h^k(s_h^k))^{2^{m+1}} - (P_{s_h^k,a_h^k}(V_{h+1}^k)^{2^m})^2\right)I_h^k$$

$$\leqslant \check{M}_{m+1} + O(d\log^5(dHK)) + \sum_{k,h}\left((V_h^k(s_h^k))^{2^{m+1}} - (P_{s_h^k,a_h^k}(V_{h+1}^k)^{2^m})^2\right)I_h^k$$

$$\leqslant \check{M}_{m+1} + O(d\log^5(dHK)) + \sum_{k,h}\left((V_h^k(s_h^k))^{2^{m+1}} - (P_{s_h^k,a_h^k}V_{h+1}^k)^{2^{m+1}}\right)$$

$$\leqslant \check{M}_{m+1} + O(d\log^5(dHK)) + 2^{m+1}\sum_{k,h}I_h^k \cdot \max\{V_h^k(s_h^k) - P_{s_h^k,a_h^k}V_{h+1}^k, 0\}$$

$$\leqslant \check{M}_{m+1} + O(d\log^5(dHK)) + 2^{m+1}\sum_{k,h}I_h^k\left(r(s_h^k,a_h^k) + \max_{\boldsymbol{\theta}\in\Theta_k}\boldsymbol{x}_{k,h}^0(\boldsymbol{\theta} - \boldsymbol{\theta}^*)\right)$$

$$\leqslant \check{M}_{m+1} + O(d\log^5(dHK)) + 2^{m+1}(K + \check{R}_0). \tag{57}$$

Finally, by (56), (57) and Lemma 24, we have

$$\check{R}_m \leqslant O\left(d^4\sqrt{(\check{M}_{m+1} + O(d\log^5(dHK)) + 2^{m+1}(K + \check{R}_0) + \check{R}_{m+1} + 2\check{R}_m)\iota\log^7(dHK)} + d^6\iota\log^5(dHK)\right)$$

$$\leqslant O\left(d^4\sqrt{(\check{M}_{m+1} + 2^{m+1}(K + \check{R}_0) + \check{R}_{m+1} + \check{R}_m)\iota\log^7(dHK)} + d^6\iota\log^5(dHK)\right), \tag{58}$$

which proves the first part of the lemma. By (54) and (57), we have

$$\left|\check{M}_m\right| \leqslant O\left(\sqrt{(\check{M}_{m+1} + O(d\log^5(dHK)) + 2^{m+1}(K + \check{R}_0))\log(1/\delta)} + \log(1/\delta)\right), \quad (59)$$

which proves the second part of the lemma. $\qquad\square$

### E.7   Proof of Lemma 23

*Proof.* Let $b_m = \check{R}_m + |\check{M}_m|$. By (58) and (59), we can bound $b_m$ recursively as

$$b_m \leqslant O\left(\sqrt{d^9 \log^5(Td)\log\frac{1}{\delta}}\sqrt{b_m + b_{m+1} + 2^{m+1}(K + \check{R}_0)} + d^7 \log^6(Td)\log\frac{1}{\delta}\right). \quad (60)$$

Note that $b_m \leqslant 2KH$ for $m \in \Lambda_1$. By Lemma 12 with parameters

$$\lambda_1 = 2KH, \quad \lambda_2 = \sqrt{d^9 \log^5(Td)\log(1/\delta)}, \quad \lambda_3 = K + \check{R}_0, \quad \lambda_4 = d^7 \log^6(Td)\log(1/\delta),$$

we obtain that

$$\check{R}_0 \leqslant b_0 \leqslant O\left(\sqrt{d^9(K + \check{R}_0)\log^5(Td)\log(1/\delta)} + d^9 \log^6(Td)\log(1/\delta)\right),$$

which implies

$$b_0 \leqslant O\left(d^{4.5}\sqrt{K\log^5(Td)\log(1/\delta)} + d^9 \log^6(Td)\log(1/\delta)\right)$$

and completes the proof. $\qquad\square$