# OpenReview forum: "Improved Variance-Aware Confidence Sets for Linear Bandits and Linear Mixture MDP"
_NeurIPS.cc/2021/Conference — NeurIPS 2021 Poster_

### Official Review · Reviewer_SrYX · 2021-07-09

**Rating:** 7
**Confidence:** 4

**Summary:**

This paper proposes a new design of the confidence set in the linear bandit problem and linear mixture RL problem. The new confidence set leads to a different regret bound, that has worse dependence on the dimension but only logarithmic dependence on the horizon length. This new design of confidence sets is the main contribution, which composes the convex potential lemma and the peeling argument.

**Limitations And Societal Impact:**

Yes. The authors clearly state the limitations of the current method and discuss the possible extensions or improvements.

**Main Review:**

A good paper that presents new ways to construct variance-aware confidence sets. The peeling argument is interesting, along with the convex potential lemma. The results have logarithmic dependence on the horizon, which is strong. Although the $d^{4.5}$ dependence is not favorable and it looks not easy to reduce the dependence from the convex potential lemma. I only carefully checked the proof for the linear bandit part. There is no obvious technical bug. But I would suggest the authors re-organize the proof to make the structure more clear. Currently, the proof of some lemma is delayed into later sections, while others are shown directly after the statement of the lemma.

For the clarity of the presentation, I suggest the authors write a more detailed proof sketch for the bandit problem, as it already contains most contributions of this work. It would help a lot to show the reader how the confidence sets are constructed from the idea of dividing norm layers.





**Time Spent Reviewing:**

4

---

> ### Author Response · Authors · 2021-08-09
> **Response to Reviewer SrYX**
>
> Thanks for the positive review. We will improve the structure, add a roadmap before stepping into the main algorithm and explain the algorithm in more detail.

---

### Official Review · Reviewer_mxmq · 2021-07-14

**Rating:** 7
**Confidence:** 4

**Summary:**

This paper proposes a new bandit and MDP algorithms for minimizing regret. The bandit algorithm enjoys a variance dependent bound, and the MDP algorithm (linear mixture) enjoys a horizon independent regret bound. The main technical novelty is a novel confidence set construction and a new convex potential lemma. The bandit regret bound has a worse scaling with dimensionality than the standard algorithms but can be significantly smaller depending on the variances.


**Ethical Concerns:**

no concerns.

**Limitations And Societal Impact:**

yes

**Main Review:**


[main review]

Originality 5/5: the solution is very novel.

Quality 4/5:

Clarity 4/5: The paper is mostly clear.

Significance 4/5: Although the algorithms are not computationally tractable, showing that such a variance dependent bound is possible is quite important.

There is a lot to think about the solution in this paper, and I did enjoy reading and thinking about this problem closely. My understanding is that the clipping is achieving some sort of robust estimator so that the confidence bound effectively does not have the usual Berstein-style lower-order term. The difference from something like Cantoni estimator is that  there is no explicit estimator that is constructed.

It would be helpful to compare with other variance dependent confidence sets. Here are two main differences from "Instance-Wise Minimax-Optimal Algorithms for Logistic Bandits" by Abeille, Faury, and Calauzenes:

* Abeille et al. use the model where the mean reward is $\mu(x\theta)$ rather than $x\theta$.
* Abeille's reward model is Bernoulli, so the rewards are necessarily binary. Constructing confidence bounds is EASIER for this Bernoulli case because the variance is a function of the mean. The authors are discussing something more general where the variance can be arbitrarily small (and always not larger than the variance of the Bernoulli case, when the rewards are scaled to [0,1]).

Adding comparison to logistic bandits may clarify the main point of the paper clearly.

A related comment on L236 : actually, Abeille et al. show a regret bound of $\sqrt{\sigma_*^2 K}$ where $\sigma^2$ is the variance of the best arm (of course, they assume a fixed arm set). This is worth being discussed.

For Theorem 3, it seems like the effect of epsilon-net is not analyzed? For example, Eq (33) is not true for $\theta_k$ because there is no guarantee that \mu_k is exactly one of the members in $\mathcal{B}$, no?

One main question that bothers me is the form of the confidence set. Do the authors think the current confidence set is tight? It is not clear to me whether the loose factor on the dimensionality comes from the confidence set construction or from the analysis (including the new potential lemma). I would really like to have some more understanding on this.

Also, if we do not have the issue of the lower order term from the Bernstein inequality, can we achieve $d\sqrt{\sum_k \sigma_k^2}$ easily? Just trying to see what is the source of looseness..

Remark 1: I am curious to know details about this. When you clip the subgaussian noise, there is bias that gets introduced (because after clipping it is not zero-mean anymore), and this cumulates and may hinder tight concentration bounds. That is, one cannot treat clipped subgaussian noise as if it was a bounded noise. How would you handle this?

[minor comments]

* L86: Faury et al. is not related to this the discussion here as it assumes the logistic model and it will have a different definition of  $\Lambda_{k-1}$.
* L118: "underlying 1 coefficient θ^*" => I did not understand what that "1" stands for.
* L109: the notation B_2^d(1) is used before being defined (L177)
* The proof of Theorem 4 appears in a weird way. Section D.2 is a large section, and the proof of Theorem 4 appears at a random location of L725. It was hard to navigate through. (or, did you mean to say 'proof of theorem 3' for the section D.2?)
* display after 553: 2/delta should be 4/delta?
* L582: the closing parenthesis } is in a wrong location.
* Q on section B: L486 and L591 should have $\\|\theta^* - \hat\theta\\|_{\Lambda_K}$ rather than $|\theta^* - \hat\theta|$
* notation mismatch: L185 says r_k is the reward but in the algorithm 1 y_k is the reward.
* The display after L741: I think there must be some typos here as it does not lead to (34). I've confirmed that (34) is correct, though.

----

**after rebuttal**

I am satisfied with the rebuttal. Thanks for sharing your thoughts on various questions I have raised. Please spend some time on clarifying, fixing typos, and making it complete (e.g., analyzing the effects from the $\epsilon$-net). It will make your great work even greater.

**Time Spent Reviewing:**

10

---

> ### Author Response · Authors · 2021-08-09
> **Response to Reviewer mxmq**
>
> Thanks for the positive review.  Please find our responses to your comments below.
>
> 1. [Abeille et al]:
> We will mention the difference with the confidence set in [Abeille et al]. Thanks for mentioning this paper!
>
> 2. Eq(33):
> The error could be covered by the term $l_{j}\iota$. That is $l_j^2+ \frac{1}{HK}\leq l_j\iota$ when $\iota \geq 4$.
>
> 3. Tightness of the current confidence set:
> We believe the current confidence set is almost tight. Actually, the confidence set is built by the one-dimensional confidence set in $(HK)^{d}$ directions. If we can improve the bound in Lemma1 to $\tilde{O}(d)$, then the regret bound would be $\tilde{O}(d^{3/2}\sqrt{\sum_i \sigma_i^2}+d^2)$. To estimate the variance we need to pay another $\sqrt{d}$ factor (see eq(27)).
> If we know the variance, the final regret bound could be $\tilde{O}(d\sqrt{\sum_i \sigma_i^2}+d^2)$, which is optimal up to log factors. The main idea is that we first divide the steps into different groups according to the magnitude of the variance using the doubling trick. Then we can regard the variance as a known constant. In this case, $\Psi$ is bounded by $O(\Phi \sigma^2)$. Plugging this into eq(27), it suffices to bound $\sum_k \frac{x_k\mu_k}{\sqrt{\Phi_k}}$. For the lower order term: When the lower order term from the Bernstein inequality is ignored, the variance-aware elliptical confidence set in [Zhou et al. 2020a] is strong enough to reconstruct the current result. The derivation is much simpler following previous methods in the linear bandit problem.
>
> 4. For sub-gaussian noise:
> We use $L:= C\log(1/\delta’)$ as the threshold for clipping. Since the mean reward is bounded in [-1,1], we have that with probability $\delta’$, the final reward will not escape [-L, L]. With the clipped reward as an estimator, the shift is bounded by $\tilde{O}(\delta’ L)$, which leads to at most $poly(KHL)$ regret. By choosing $\delta= \frac{\delta’}{poly(KH)}$, the regret due to the shift is bounded.
>
>
>
> Minor comments:
> Thanks for pointing out these issues! We will fix typos accordingly.
>
> L86: We mentioned Faury et al. since they also use variance-aware confidence set. It is natural to transform their method to the linear bandit case.
>
> L118: Sorry for the typo. The “1” is redundant.
>
> Section D: Yes it’s Theorem 3. We will add another subsection for the proof of Theorem 3.
>
> L553: Yes. It should be $4/\delta$.

---

### Official Review · Reviewer_hVCh · 2021-07-17

**Rating:** 6
**Confidence:** 4

**Summary:**

This paper studies the regret bound of linear bandits and linear mixture MDP problems. The authors develop algorithms with variance-aware confidence sets and achieve a data-dependent regret bound.

**Limitations And Societal Impact:**

yes

**Main Review:**

Strengths

1. For linear bandits, the regret bound only scales with the variance and the dimension. If the variances are very small, the bound can be significantly smaller than $\sqrt{K}$, where $K$ is the number of rounds.

2. For the linear mixture MDP in the linear approximation settings, this work introduces an algorithm that is only scaling logarithmically with the horizon $H$, which is important when $H$ is relatively large.

3. The algorithms do not need the information of the true variance.

4. The reviewer thinks that the technical ideas developed in the paper would probably be of interest to the community.

Weaknesses

1. Is it possible to remove the assumption $\sum_h^H r_h \le 1$ (Assumption 2) that is not required in work [1]? This assumption is not generally true in practice. Without this assumption, we have $\sum_h^H r_h \le H$, and then the regret bound may depend more on $H$ than the current result. The reviewer thinks that the difference between the result in the paper and the result in [1] should be mentioned.

2. Although this work's strength is in its theoretical development, however, the supplemental material is badly structured, which is difficult to follow. The reviewer thinks that a proof roadmap should be given. Also, the titles are misleading. For example, in the proof of Theorem 4 in the supplemental material, Line 722-724, it says that Theorem 3 is proved. And in Line 725, some proof jumps in without any reference.

3. The algorithms presented are computationally inefficient as mentioned in the discussion part of the paper. The reviewer thinks that the computational complexity should be analyzed with more detail.

Other comments:

1. Page 2 Line 40: The reviewer suggests that the full name of ``MVP'' be presented.

2. Page 5 Line 189-193 Remark 1: Does $T$ represent the number of rounds? The notation is confusing since $K$ is used for the number of rounds in the rest of the paper.


[1] Zhou, Dongruo, Quanquan Gu, and Csaba Szepesvari. "Nearly minimax optimal reinforcement learning for linear mixture markov decision processes." arXiv preprint arXiv:2012.08507 (2020).



**Time Spent Reviewing:**

16

---

> ### Author Response · Authors · 2021-08-09
> **Response to Reviewer hVCh**
>
> Thanks for carefully reviewing. We will improve the presentation and fix the typos. Please find our response to your comments below:
>
> 1. About the reward-scaling assumption:
> We note that when comparing with [Zhou et al 2020a], we rescaled their bound (line 161) under the assumption that every reward $r_h \le 1/H$ in order to make the total reward $\sum_{h} r_h \le 1$ as in Assumption 2. The bound stated in their paper is $H$ times larger than what we stated in line 161. If the reward-scaling assumption is $r_h \le 1$ for every $h$. Our bound will scale up by an $H$ factor.
> Assumption 2 actually is a more realistic assumption than $r_h \le 1/H$ for every $h$. See line 209  - 212, and [Jiang and Agarwal 2018], [Wang et al. 2020a], [Zhang et al. 2020a] for discussions about reward scaling.
> We will add more clarifications. Thanks for asking.
>
> 2. About the computation complexity:
> the major computational cost is enumerating over $\mathcal{B}$, which is $O(T^{O(d)})$. As we have discussed in line 330-342, an interesting future direction is to develop a computationally efficient algorithm for the problems considered in this paper.
>
> 3. Line 189-193 Remark 1:
> Yes, $T$ is $K$. Sorry for the typo.

---

### Official Review · Reviewer_hw6t · 2021-07-23

**Rating:** 7
**Confidence:** 3

**Summary:**

This paper discusses an interesting topic, the variance awareness approach in online learning and decision making (e.g., bandit and linear MDP).  Two algorithms VOFUL (Variance-awareness OFUL) and VARLin (Variance-awareness RL with linear model) are proposed. By adaptive shrink the feasible parameter set \Theta_{k} based on the estimated variance, the authors theoretically show the regrets upper  bounds as \tilde{O}(poly(d)\sqrt{1+\sum_{k}\sigma^2_k} and \tilde{O}(ploy(d,log H)\sqrt{K}) for bandits and linear MDP respectively. Moreover, several technical ideas that may be of independent interest are proposed: the peeling technique to both the input norm and the variance magnitude, 2) a recursion-based estimator for the variance, and 3) a new convex potential lemma that generalizes the seminal elliptical potential lemma.

**Limitations And Societal Impact:**

1) Line 83-91, the example is kind of written in a misleading way. Since we require sigma^2<<1 and d = 1, how does the first inequality on Line 89 go through?

2) Since the major contribution of this paper is on regrets in theory, it ok to have a huge computation cost to run the algorithms. And in section 6, the authors also discuss the computation issue and argue that many statistically efficient algorithms are computationally inefficient.   It would be better to add some discussions in the main paper to point out some modifications on the algorithm to improve the computation efficiency.

**After engaging in the discussion, I think the paper discusses an interesting topic and contains non-trivial results. The authors have already addressed my questions. Therefore, I would like to change my score to accept this paper.

**Main Review:**

Originality: This paper proposed two new noise awareness OFUL type algorithms, VOFUL and  VARLin for the linear bandit and linear RL respectively. Both Algorithms first estimate the error terms, use them to shrink the parameter‘s feasible set \Theta, and finally run the unpenalized regression. For linear bandit, the proposed algorithm reaches O(poly(d)\sqrt{1+\sum_{k}\sigma^2_k}) data-dependent upper bound, where k is the time step. For linear RL, O(poly(d,\log H)\sqrt{K}) bound is achieved, where H is the planning horizon. This result improves the dependence in H from linear to logarithmic compared to the literature, e.g., Zhou et al. [2020a]. Moreover, several technical results that may be of independent interest are proposed.

Quality:  The proposed model reaches promising H dependence for linear RL settings. This novel result improves the literature. However, for the linear bandit case, I’m a little bit concerned about the author's claim on "no explicit dependence on K". In fact, if the variance \sigma_k is a constant over time, the bound in this paper becomes O(poly(d)\sqrt{1+k\sigma^2}), which contains the k dependence as long as sigma is not  0. Moreover, in Theorem 1 and Theorem 2 in [1], the O(poly(d)\sigma\sqrt{K}) dependence has already be achieved. Based on the current draft, the author's claim on the bandit may not be well supported.

Clarity: The presentation of this paper is ok but it can be certainly further improved. For example, the authors present two algorithms with relatively complex structures, more discussions or remarks on how those algorithms work, and why they allow gaining better regret bound will be helpful.

Significance: This paper discusses an interesting question that we may use adaptive procedures with noise awareness to gain better regrets. In general, it is an exciting idea since in the real world the online problems involving distribution drifting are very common. If we can adaptively tune the algorithm instead of always assuming the worst-case parameters to ensure convergence, the numerical performance can be very promising. The regrets bound for linear RL is new and improves the results in the literature. However, I'm a little bit doubt about the novelty claim  "no explicit dependence on K" in bandit settings and the authors may need to make further justification. Moreover, the d dependence is at least d^4.5, which is significantly higher than O(d\sqrt{T}) or O(\sqrt{dT}) bounds in the bandit literature.


[1] Li, Lihong, Yu Lu, and Dengyong Zhou. "Provably optimal algorithms for generalized linear contextual bandits." International Conference on Machine Learning. PMLR, 2017.


**Time Spent Reviewing:**

10

---

> ### Author Response · Authors · 2021-08-09
> **Response to Reviewer hw6t**
>
> Thanks for your careful review. We will add some remarks to explain the algorithm to make it easier to read. Please find our responses to your comments below:
>
> 1. Concern about “no explicit dependence on $K$”: As we have discussed in line 25-36, it is well-known that the minimax optimal regret for linear bandits is $\tilde{\Theta}(d\sqrt{K})$ in the worst-case (when variance is constant as you suggested). Our goal is to obtain a regret bound that is adaptive to the variance magnitude so in the noiseless case it becomes constant-type bound and in the worst case it becomes \sqrt{K}-type bound (see line 35-36). To achieve this, there can be no explicit polynomial dependence on $K$ in the regret bound because otherwise, the regret bound cannot be constant-type in the noiseless case. We will further clarify this point in the final version.
>
>
> 2. Line 83-91: In the first inequality, we use the fact that $\sigma\sqrt{d}$ is non-negative. Actually, what really matters is that the term $1$ is unavoidable when using previous confidence sets.

---

> ### Author Response · Authors · 2021-08-20
> **HAS OUR RESPONSE ADDRESSED YOUR CONCERNS?**
>
> Hello reviewer hw6t, we would be grateful if you can confirm whether our response has addressed your concerns, and let us know if any issues remain. To recap our response, we:
>
> •	Clarified what we meant by "explicit dependence on $K$".
>
> •	Explained why the first inequality in line 83-91 holds.

---

> > ### Comment · Reviewer_hw6t · 2021-08-21
> > **Thanks for the response**
> >
> > Dear authors,
> >
> > Your response has already addressed my questions, and I'm good with it.
> >
> > Thanks

---

> > > ### Author Response · Authors · 2021-08-23
> > > **Thanks for your reply!**
> > >
> > > Thanks for your reply! Please let us know if you have any other questions.

---

### Decision · Program_Chairs · 2021-09-27

**Decision:**

Accept (Poster)

**Comment:**

This paper considers the problem of adapting to noise variance in linear bandits and reinforcement learning (linear mixture MDPs). The reviewers agree that the main technical result in the paper (a new confidence set construction which is adaptive to variance) is interesting and highly non-trivial. This is a timely result, and I am confident this technique will find broader use. In addition, there are many interesting directions (e.g., improving computational efficiency) for future work.

While the reviewers felt that the paper is generally well-written, the authors are encouraged to incorporate their suggestions to improve the clarity and organization of the main body and appendix.